# Solving Imperfect-Recall Games via Sum-of-Squares Optimization

## Abstract

Extensive-form games (EFGs) provide a powerful framework for modeling sequential decision making, capturing strategic interaction under imperfect information, chance events, and temporal structure. Most positive algorithmic and theoretical results for EFGs assume perfect recall, where players remember all past information and actions. We study the increasingly relevant setting of imperfect-recall EFGs (IREFGs), where players may forget parts of their history or previously acquired information, and where equilibrium computation is provably hard. We propose sum-of-squares (SOS) hierarchies for computing ex-ante optimal strategies in single-player IREFGs and Nash equilibria in multi-player IREFGs, working over behavioral strategies. Our theoretical results show that (i) these hierarchies converge asymptotically, (ii) under genericity assumptions, the convergence is finite, and (iii) in single-player non-absentminded IREFGs, convergence occurs at a finite level determined by the number of information sets. Finally, we introduce the new classes of (SOS)-concave and (SOS)-monotone IREFGs, and show that in the single-player setting the SOS hierarchy converges at the first level, enabling equilibrium computation with a single semidefinite program (SDP).

## 1 Introduction

Extensive-form games (EFGs) are a central framework for modelling sequential decision-making under imperfect information, with important applications in economics, operations research, and artificial intelligence (Osborne & Rubinstein, 1994; Fudenberg & Tirole, 1991). An EFG represents interactions on a tree in which players (or chance) take turns choosing actions. Each sequence of actions defines a history that either terminates at a leaf with associated payoffs or leads to another decision point. This representation is also able to capture imperfect information, where players cannot distinguish between different nodes. EFGs have been used extensively in modern AI, particularly in solving large-scale imperfect information games (Brown & Sandholm, 2018; Bowling et al., 2015; Moravčík et al., 2017; Bakhtin et al., 2022).

A key distinction between classes of EFGs is the ability of players to retain memory. In perfect-recall EFGs, players remember all past information sets and actions, whereas in imperfect-recall EFGs (IREFGs) some information may be lost, such as forgetting which action was taken or even whether a state has been visited before. Recent work has increasingly focused on IREFGs, since they capture a more realistic model of decision-making. IREFGs are used to abstractify large games, letting agents ignore strategically unimportant details (Waugh et al., 2009; Ganzfried & Sandholm, 2014; Brown et al., 2015; Čermák et al., 2017a). They also provide a natural representation for bounded rationality (Lambert et al., 2019), for describing teams of cooperating agents as a single imperfect-recall decision maker (Von Stengel & Koller, 1997; Celli & Gatti, 2018; Zhang et al., 2022), and for designing privacy-preserving or data-restricted agents (Conitzer, 2019; Conitzer & Oesterheld, 2023).

EFGs admit two principal strategy formalisms: mixed strategies, defined as distributions over pure strategies that select an action everywhere in the game tree, and behavioral strategies, which randomize between actions independently at each information set. The canonical solution concept is a Nash equilibrium (NE), a strategy profile from which no player can unilaterally deviate. Perfect recall plays a crucial role in the theoretical guarantees of EFGs: by Kuhn's theorem (Kuhn, 1953), mixed and behavioral strategies are equivalent in perfect recall EFGs, implying the existence of behavioral NEs

and enabling efficient algorithms, such as polynomial-time methods for solving two-player zero-sum EFGs (Koller & Megiddo, 1992).

The imperfect-recall setting is far more challenging. The equivalence between mixed and behavioral strategies breaks down, and mixed NE still exist but may be unimplementable since they require players to have perfect memory of the game states. Though behavioral NE are a more natural solution concept, they might not exist, even in two-player zero-sum IREFGs (Wichardt, 2008). Computationally, solving two-player zero-sum IREFGs is NP-hard (Koller & Megiddo, 1992), deciding whether a single-player IREFG achieves a target value is hard (Gimbert et al., 2020; Tewolde et al., 2023), even relaxed equilibrium notions such as EDT and CDT equilibria are hard to compute (Tewolde et al., 2023), and deciding whether a behavioral NE exists is hard (Tewolde et al., 2024).

Many recent hardness results for IREFGs (Gimbert et al., 2020; Tewolde et al., 2023; 2024) rely on the folklore observation that any IREFG over behavioral strategies can be expressed as a polynomial game over a product of simplices (Piccione & Rubinstein, 1997). Importantly, the transformation from an EFG to its polynomial game representation can be carried out in polynomial time. Polynomial games form a key subclass of continuous games, originally studied by Dresher et al. (1950), in which each player's utility is a polynomial in the decision variables of all players.

On the other hand, there exists well-developed machinery for handling single- and multi-player optimization problems where utilities are polynomial and strategy sets are semi-algebraic. In the single-player case, the Moment-SOS hierarchy (Lasserre, 2001; Parrilo, 2000) applies to polynomial optimization problems of the form $\max f(x)$ subject to $x \in \mathcal{X}$, where $f$ is a polynomial and $\mathcal{X}$ is defined by polynomial equalities and inequalities. Although such problems are NP-hard in general (Motzkin & Straus, 1965), the hierarchy produces a sequence of semidefinite relaxations whose optimal values converge to the true optimum under mild assumptions. At relaxation level $d$, it seeks the smallest scalar $t$ such that $t - f(x)$ admits a sum-of-squares (SOS) certificate of nonnegativity on $\mathcal{X}$, a condition expressible as a semidefinite program (SDP) of polynomial size. Extensions of the Moment-SOS hierarchy have also been developed for the multi-player polynomial optimization setting, applied to polynomially representable supersets of the equilibrium set (Nie & Tang, 2024).

Thus far, the link between IREFGs and polynomial optimization has mostly served to prove hardness results, but we instead use it as a foundation for positive algorithmic results. Specifically, we ask:

> *What guarantees does the Moment-SOS hierarchy provide for solving IREFGs?*
> *Are there structured subclasses of IREFGs where it enables tractable computation?*

**Contributions.** By bringing these tools to imperfect-recall games, we obtain a general, provably convergent framework for computing behavioral equilibria in IREFGs. Moreover, we identify structural conditions under which convergence occurs at finite or low levels of the hierarchy, leading to efficient algorithms for broad and natural subclasses. Specifically, our main contributions are:

- For *single-player IREFGs*, the Moment-SOS hierarchy converges asymptotically to the ex-ante optimal value. Under a *genericity* assumption, we show that convergence is instead finite for almost all games. Moreover, in non-absentminded IREFGs (NAM-IREFGs), we show that exact convergence occurs at a level of the hierarchy depending on the number of infosets in the game.

- For *multi-player IREFGs*, we adapt an approach proposed in recent work by Nie & Tang (2024). The method requires multiple instantiations of the Moment-SOS hierarchy and converges asymptotically to a behavioral NE, certifying non-existence otherwise. As with the single-player case, for almost all games the convergence is finite. In NAM-IREFGs, we show that a variant of the method generically converges in finite time, requiring only a single instantiation of the hierarchy.

- We define the tractable subclasses of concave and monotone IREFGs, and SOS-certifiable counterparts thereof. In single-player SOS-concave/SOS-monotone IREFGs, we show that the Moment-SOS hierarchy converges at the *first* level, enabling computation of ex-ante optima with a single SDP.

**Related work in IREFGs.** Existing positive results for IREFGs typically rely on restrictions that admit tractable perfect-recall refinements. A-loss recall games (Kaneko & Kline, 1995; Kline, 2002) limit forgetting to past actions, enabling sufficient conditions for behavioral NE to exist and approximation methods in the two-player zero-sum case (Čermák, 2018). NAM-IREFGs, wherein

players always remember previously encountered decision points, can be transformed into equivalent (though exponentially larger) A-loss recall games (Gimbert et al., 2025). Finally, chance-relaxed skew well-formed games are a subclass of IREFGs where counterfactual regret minimization still provably minimizes regret (Lanctot et al., 2012; Kroer & Sandholm, 2016).

## 2 PRELIMINARIES

### 2.1 IMPERFECT RECALL EXTENSIVE-FORM GAMES (IREFGS)

We first define extensive-form games of imperfect recall. For a more thorough review of standard concepts in EFGs, the reader is referred to Fudenberg & Tirole (1991); Osborne & Rubinstein (1994).

An $n$-player extensive form game $\mathscr{G}$ is a tuple $\mathscr{G} := \langle \mathcal{H}, \mathcal{A}, \mathcal{Z}, \rho, \mathcal{I} \rangle$ where:

- The set $\mathcal{H}$ denotes the states of the game which are decision points for the players. The states $\pi \in \mathcal{H}$ form a tree rooted at an initial state $r \in \mathcal{H}$. We denote terminal nodes in $\mathcal{H}$ by $\mathcal{Z}$. Each nonterminal state $\pi \in \mathcal{H} \setminus \mathcal{Z}$ is associated with a set of *available actions* $\mathcal{A}_\pi$.
- Given $\mathcal{N} = \{1, \ldots, n\}$, the set $\mathcal{N} \cup \{c\}$ denotes the $n+1$ players of the game. Each state $\pi \in \mathcal{H}$ admits a label $\text{Label}(\pi) \in \mathcal{N} \cup \{c\}$ which denotes the *acting player* at state $\pi$. The letter $c$ denotes a *chance player*, representing exogenous stochasticity. $\mathcal{H}_i \subseteq \mathcal{H}$ denotes the states $\pi \in \mathcal{H}$ with $\text{Label}(\pi) = i$. Each chance node $\pi \in \mathcal{H}_c$ is associated with a fixed distribution $\mathbb{P}_c(\cdot | \pi)$ over $\mathcal{A}_\pi$, denoting the distribution over actions chosen by the chance player at each node.
- For each $i \in \mathcal{N}$, payoff function $\rho_i : \mathcal{Z} \to \mathbb{R}$ specifies the payoff that player $i$ receives if the game ends at terminal state $z \in \mathcal{Z}$.
- The game states $\mathcal{H}$ are partitioned into *information sets* (also called infosets) ascribed to each player, namely $\mathcal{I}_i \in (\mathcal{I}_1, \ldots, \mathcal{I}_n)$. Each information set $I \in \mathcal{I}_i$ encodes groups of nodes that the acting player $i$ cannot distinguish between, so if $\pi_1, \pi_2 \in I$, then $\mathcal{A}_{\pi_1} = \mathcal{A}_{\pi_2}$. We let $\mathcal{A}_I$ denote the shared action set of infoset $I$.
- For notational convenience, we ascribe a singleton information set to each chance node and define $\mathcal{I}_c$ as the collection of these chance node infosets. For each non-terminal node $\pi \in \mathcal{H} \setminus \mathcal{Z}$, we thus define $I_{\pi \in (\mathcal{I}_1, \ldots, \mathcal{I}_n) \cup \mathcal{I}_c}$ to be the infoset it belongs to.

**Memory.** There are two key distinctions regarding players' memory. A game has *perfect recall* if no player ever forgets their past history, namely, the sequence of information sets visited, actions taken within those information sets, and any information acquired along the way. Formally, for any information set $I \in \mathcal{I}_i$ and any two nodes $\pi_1, \pi_2 \in I$, the sequence of Player $i$'s actions from the root $r$ to $\pi_1$ must coincide with the sequence from $r$ to $\pi_2$; otherwise, the player could distinguish between the two nodes. A game is said to have perfect recall if this property holds for all players; otherwise, it is said to have the *imperfect recall* property.

**Strategy formalism.** A *pure strategy* specifies a deterministic action at every information set of a player. A *mixed strategy* is a probability distribution over pure strategies. However, mixed strategies require players to coordinate their actions across information sets, which implicitly assumes memory and therefore conflicts with the imperfect-recall setting. For this reason, IREFGs are most naturally analyzed using *behavioral strategies*, which specify independent randomizations at each information set. Formally, for any information set $I \in \mathcal{I}_i$, let $\Delta(\mathcal{A}_I)$ denote the simplex of probability distributions over the available actions $\mathcal{A}_I$. A behavioral strategy for player $i$ is a mapping $\mu_i : \mathcal{I}_i \to \bigcup_{I \in \mathcal{I}_i} \Delta(\mathcal{A}_I)$, assigning to each information set $I$ a distribution $\mu_i(\cdot \mid I) \in \Delta(\mathcal{A}_I)$. The joint behavioral strategy profile for all players is denoted by $\mu := (\mu_i)_{i \in \mathcal{N}}$.

Kuhn's theorem (Kuhn, 1953) establishes that, in games with perfect recall, behavioral and mixed strategies are outcome-equivalent—that is, they induce the same probability distribution over terminal histories. This equivalence no longer holds in the imperfect-recall setting, making behavioral strategies the canonical choice for IREFGs.

Going forward, we establish some additional notational conventions regarding behavioral strategies. We use $(\mu_i, \mu_{-i})$ to describe the influence of player $i$ on $\mu$, where $\mu_{-i}$ collects all components except player $i$. Let $\ell_i$ denote the number of infosets of player $i$, i.e. $\ell_i := |\mathcal{I}_i|$, and $m_i^j$ denote the number of actions in a given infoset $I_i^j \in \mathcal{I}_i$ of player $i$, i.e. $m_i^j := |\mathcal{A}_{I_i^j}|$. The strategy set of player $i$ over all

their infosets can be written as a Cartesian product of simplices: $\mathcal{S}_i := \bigtimes_{j=1}^{\ell_i} \Delta^{m_i^j - 1}$. Finally, the strategy set over all players is $\mathcal{S} := \bigtimes_{i=1}^{n} \mathcal{S}_i$. In the single-player case, we set $n = 1$ and drop the player index $i$ for clarity.

**Solution concepts.** The expected utility of player $i \in \mathcal{N}$ following (joint) behavioral strategy $\mu$ is $u_i(\mu) := \sum_{z \in \mathcal{Z}} \mathbb{P}(z|\mu, r) \cdot \rho_i(z)$, where $\mathbb{P}(z|\mu, r)$ is the probability that leaf $z \in \mathcal{Z}$ is reached from root $r$ following joint behavioral strategy $\mu$. In a single-player IREFG $\mathscr{G}$, a behavioral strategy $\mu^*$ is said to be *ex-ante optimal* if it is a solution to the following optimization problem:

$$\max_{\mu} \ u(\mu) \quad \text{s.t.} \ \mu \in \mathcal{S}. \tag{1}$$

In these games, $\mathcal{S}$ is compact and $u$ is continuous, so $\arg\max_{\mathcal{S}} u \neq \emptyset$ (i.e. ex-ante optima always exist). In a multi-player IREFG $\mathscr{G}$, a joint behavioral strategy $\mu^* \in \mathcal{S}$ is a (behavioral) Nash equilibrium if for every player $i \in \mathcal{N}$, we have that:

$$u_i(\mu^*) \ \geq \ u_i(\mu_i, \mu^*_{-i}), \quad \forall \mu_i \in \mathcal{S}_i, \tag{2}$$

i.e., no player can profitably deviate from $\mu^*$ to any other behavioral strategy. Importantly, behavioral Nash equilibria need not exist in IREFGs; a counterexample is given in Wichardt (2008) (cf. Figure 2.1 (a)). Moreover, even deciding whether one exists is hard (Tewolde et al., 2024).

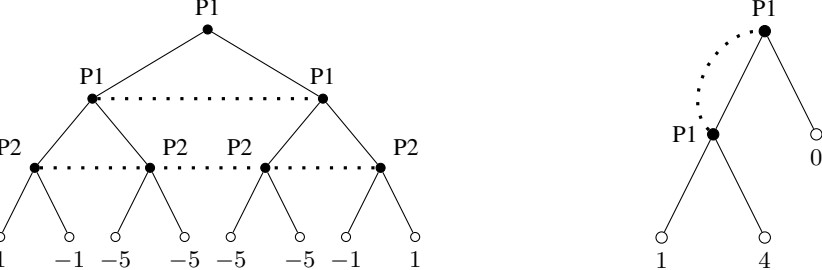

Figure 2.1: (a) A two-player zero-sum IREFG with no NE; (b) the single-player absentminded taxi driver IREFG. Dotted lines denote infosets. In (b), P1 cannot distinguish between nodes in the same history, so behavioral strategies (e.g., *Left* w.p. $x$, *Right* w.p. $1 - x$) yield expected utility $u(x) = x^2 + 4x(1 - x)$, a polynomial in the behavioral strategy space.

## 2.2 THE MOMENT/SUM-OF-SQUARES HIERARCHY

We will utilize ideas from the polynomial optimization literature to solve IREFGs. We direct the reader to Lasserre (2009b); Laurent (2009) for more thorough discussions of these techniques. Here we give only a brief overview of the Moment-SOS hierarchy tailored to our purposes; a step-by-step derivation in our setting is provided in Appendix D.

We now introduce the Moment-SOS hierarchy. Consider the polynomial optimization problem:

$$f^* := \max_x \{f(x) : x \in \mathcal{X}\}, \tag{3}$$

where the feasible region is a basic semi-algebraic set $\mathcal{X} := \big\{x \in \mathbb{R}^m : g_j(x) \geq 0, j \in [\![m_g]\!], h_k(x) = 0, k \in [\![m_h]\!]\big\}$ for two families of polynomials $g_1, \ldots, g_{m_g}, h_1, \ldots, h_{m_h} \in \mathbb{R}[x]$. Let $\alpha := (\alpha_1, \ldots, \alpha_m) \in \mathbb{N}^m$ be a multi-index, and write $x^\alpha := \prod_{i=1}^{m} x_i^{\alpha_i}$, $|\alpha| := \sum_{i=1}^{m} \alpha_i$. For two multi-indices $\alpha, \beta$, the sum $\alpha + \beta$ is taken componentwise, i.e., $(\alpha + \beta)_i = \alpha_i + \beta_i$. Denote by $\mathbb{R}[x]$ the ring of real polynomials in the variables $x := (x_1, \ldots, x_m)$, and by $\mathbb{R}[x]_d$ the vector space of polynomials of degree at most $d$ (whose dimension is $s(m) := \binom{m+d}{m}$). A sequence $y = (y_\alpha)_{\alpha \in \mathbb{N}^m} \subset \mathbb{R}$ defines a Riesz linear functional $L_y : \mathbb{R}[x] \to \mathbb{R}$, $L_y(x^\alpha) = y_\alpha$. Given a sequence $y$ and an integer $d \geq 0$, the *degree-d moment matrix* $M_d(y)$ is the symmetric matrix with rows and columns indexed by all multi-indices $\alpha, \beta$ satisfying $|\alpha|, |\beta| \leq d$, and whose entry is $M_d(y)_{\alpha,\beta} = y_{\alpha+\beta}$. For any polynomial $g(x) = \sum_\gamma g_\gamma x^\gamma$, the *degree-d localizing matrix* of $y$ with respect to $g$ is $M_d(g \star y)_{\alpha,\beta} := L_y(g(x)x^{\alpha+\beta}) = \sum_\gamma g_\gamma y_{\alpha+\beta+\gamma}$, $|\alpha|, |\beta| \leq d - \lceil \deg(g)/2 \rceil$.

Denote by $\Sigma[x] := \big\{\sigma \in \mathbb{R}[x] : \sigma(x) = \sum_{k=1}^{K} q_k(x)^2, \ q_k \in \mathbb{R}[x]\big\}$ the set of SOS polynomials and by $\Sigma[x]_d$ the set of SOS polynomials of degree at most $2d$. The quadratic module of $\mathcal{X}$ is $Q(\mathcal{X}) :=$

$\left\{\sigma_0 + \sum_{j=1}^{m_g} \sigma_j\, g_j + \sum_{k=1}^{m_h} p_k\, h_k : \sigma_0, \sigma_j \in \Sigma[x],\ p_k \in \mathbb{R}[x]\right\}$. Its truncation at degree $d$, denoted $Q_d(\mathcal{X})$, consists of those elements for which $\deg(\sigma_0) \leq 2d$, $\deg(\sigma_j\, g_j) \leq 2d$, $\deg(p_k\, h_k) \leq 2d$. Note that simplex action sets (as are standard in IREFGs) are basic semi-algebraic sets.

Let $d_f := \lceil \deg(f)/2 \rceil$, $d_g := \max_j \lceil \deg(g_j)/2 \rceil$, $d_h := \max_k \lceil \deg(h_k)/2 \rceil$, and $d_{\mathcal{X}} := \max\{d_g, d_k\}$, then $d_0 := \max\{d_f, d_{\mathcal{X}}\}$. For $d \geq d_0$, the degree-$d$ SOS relaxation of Equation (3) is

$$\begin{aligned} f_d^{\mathrm{sos}} \;=\; &\inf_{t,\,\sigma_0,\{\sigma_j\},\{p_k\}}\ t \\ &\text{s.t.}\quad t - f(x) \in Q_d(\mathcal{X}). \end{aligned} \tag{4}$$

Its dual is the degree-$d$ moment relaxation:

$$\begin{aligned} f_d^{\mathrm{mom}} \;=\; &\sup_y\ L_y(f) \\ &\text{s.t.}\quad y_0 = 1,\quad M_d(y) \succeq 0,\quad M_{d-d_{\mathcal{X}}}(g_j \star y) \succeq 0,\ \forall j, \\ &\qquad\ \ L_y(h_k\, q) = 0,\ \forall k,\ \forall q \in \mathbb{R}[x],\ \deg(h_k q) \leq 2d. \end{aligned} \tag{5}$$

It is immediate that $f_d^{\mathrm{mom}} \leq f_d^{\mathrm{sos}}$ for all $d$, and both sequences $\{f_d^{\mathrm{sos}}\}$ and $\{f_d^{\mathrm{mom}}\}$ are nonincreasing in $d$. Moreover, the hierarchy converges asymptotically under mild assumptions on the description of the semi-algebraic set, e.g., see Lasserre (2001). In addition to approximating the optimal value, the moment relaxation also allows one to extract maximizers under appropriate assumptions. It is well known that if, at some order $d$, the flatness condition $\operatorname{rank} M_s(y_d) = \operatorname{rank} M_{s-d_{\mathcal{X}}}(y_d)$ holds for some $s \leq d$, then the relaxation is exact and an optimal strategy can be extracted from $M_s(y_d)$; see Henrion & Lasserre (2005) and Appendix D.4 for details about the extraction procedure.

**Computational considerations.** The degree-$d$ moment relaxation in Equation (5) is an SDP with $\binom{m+2d}{2d}$ moment variables $y_\alpha$, $1 + m_g$ PSD constraints where $M_d(y)$ has size $\binom{m+d}{d}$ and $M_{d-d_{\mathcal{X}}}(g_j \star y)$ has size $\binom{m+d-d_j}{d-d_j}$, $1 + \sum_{k=1}^{m_h} \binom{m+2d-2d_k}{2d-2d_k}$ linear equality constraints (normalization and the $h_k$-constraints). The dual degree-$d$ SOS relaxation Equation (4) is an SDP with on the order of $1 + \binom{m+2d}{2d} + \sum_{j=1}^{m_g} \binom{m+2(d-d_j)}{2(d-d_j)} + \sum_{k=1}^{m_h} \binom{m+2(d-d_k)}{2(d-d_k)}$ scalar variables (for $t$, the coefficients of $\sigma_0, \sigma_j$, and $p_k$), $1 + m_g$ PSD constraints for the Gram matrices of $\sigma_0, \sigma_j$, of sizes $\binom{m+d}{d}$ and $\binom{m+d-d_j}{d-d_j}$ respectively, $\binom{m+2d}{2d}$ linear equality constraints from coefficient matching.

The SDP size can grow quickly with $m$ and $d$, which constrains the Moment-SOS hierarchy to polynomial optimization problems of modest dimension. Since computational efficiency is not a focus of our work, we briefly mention that several recent works have aimed to improve the scalability of SDP solving. For instance, DSOS/SDSOS relaxations trade off solution quality for better computational performance (Ahmadi & Majumdar, 2019), and recent low-rank SDP methods reduce memory and runtime by exploiting approximate low-rank structure (Monteiro et al., 2024; Han et al., 2024; 2025; Aguirre et al., 2025). Our polynomial IREFG formulation fits into this framework, so these techniques could be combined with our degree bounds to handle larger imperfect-recall benchmarks. Exploring these scalable SDP solvers is a key direction for future work.

## 3 THE LINK BETWEEN IREFGS AND POLYNOMIAL OPTIMIZATION

We begin this section the with a folklore result that bridges the study of IREFGs with polynomial optimization, originating in Piccione & Rubinstein (1997) and expanded further in Tewolde et al. (2023; 2024). A thorough exposition of this connection is given in Appendix B.

**Theorem 3.1** (Folklore). *In any IREFG, the expected utility of a player $i \in \mathcal{N}$ under a joint behavioral strategy $\mu$ is a polynomial in the entries of $\mu$. As a consequence:*

*(i) In the single-player case, computing an ex-ante optimal behavioral strategy reduces to a polynomial optimization problem over products of simplices.*

*(ii) In the multi-player case, computing a behavioral Nash equilibrium $\mu^* \in \mathcal{S}$ reduces to solving coupled polynomial optimization problems over products of simplices, i.e.,*

$$\mu_i^* \;\in\; \underset{\mu_i \in \mathcal{S}_i}{\arg\max}\ u_i(\mu_i, \mu_{-i}^*),\quad \forall i \in \mathcal{N}.$$

*Moreover, these reductions can be carried out in polynomial time.*

In view of this, while the Moment-SOS hierarchy applies directly in the single-player case, this is not the case for the multi-player setting. We next introduce two modifications that extend its applicability to the multi-player case and provide stronger convergence guarantees.

**KKT-based Moment-SOS hierarchy.** In the multi-player case, the reduction yields a polynomial game over a product of simplices. However, it is unclear how to use this, since the set of NEs admits no explicit semi-algebraic description. To apply the Moment/SOS hierarchy in this setting, we follow (Nie & Tang, 2024). At a high level, their approach defines a semi-algebraic superset of the Nash equilibria and then applies the Moment/SOS hierarchy to this larger set. This set is obtained by concatenating the KKT conditions of each individual player's optimization problem. These conditions are already polynomial in the primal and dual variables, but under appropriate conditions it is possible to reduce variables by expressing the dual multipliers as polynomials of the primal variables.

To explain this approach, we introduce some additional notation. The action space of each player $i$ is a product of simplices over its information sets, which admits a simple semi-algebraic description $\mathcal{S}_i = \{\mu_i : h_i^j(\mu) = 0, \ g_i^j(\mu) \geq 0, \ \forall j\}$, where $h_i^j(\mu) := \mathbf{1}^\top \mu_i^j - 1$, $g_{i,a}^j(\mu) := \mu_{i,a}^j \ \forall a$, $g_i^j := (g_{i,1}^j, \ldots, g_{i,m_i^j}^j)^\top$. For each player $i$ and information set $j$, we define the block gradient $w_i^j(\mu) := \nabla_{\mu_i^j} u_i(\mu) \in \mathbb{R}^{m_i^j}$. Denote by $\nu_i^j \in \mathbb{R}$ and $\lambda_i^j \in \mathbb{R}_{\geq 0}^{m_i^j}$ the Lagrange multipliers associated with the constraints $h_i^j(\mu) = 0$ and $g_i^j(\mu) \geq 0$, respectively. The KKT stationarity condition at an optimizer requires that $w_i^j(\mu) - \nu_i^j \mathbf{1} - \lambda_i^j = 0$ for all $j$. Taking the inner product with $g_i^j(\mu)$, and using both feasibility $h_i^j(\mu) = 0$ and complementary slackness $g_i^j(\mu)^\top \lambda_i^j = 0$, yields the following explicit (polynomial) expressions for the multipliers: $\nu_i^j(\mu) = g_i^j(\mu)^\top w_i^j(\mu)$ and $\lambda_i^j(\mu) = w_i^j(\mu) - \nu_i^j(\mu)\mathbf{1}$. Hence, all Lagrange multipliers are polynomials in $\mu$.

Furthermore, in our setting (i.e., product of simplices), linear independence constraint qualification (LICQ) holds at every feasible point, and hence the KKT conditions are necessary for optimality. Indeed, at any infoset block the constraints are $\mathbf{1}^\top \mu = 1$ and $\mu_a \geq 0$. Thus, the gradients of the active constraints are $\mathbf{1}$ and $e_a$ for all $a$ with $\mu_a = 0$. Since $\sum_a \mu_a = 1$ implies at least one $\mu_a > 0$, not all $e_a$ can be active, so $\{\mathbf{1}\} \cup \{e_a : \mu_a = 0\}$ is linearly independent and LICQ holds. Consequently, every NE must satisfy the joint KKT system of polynomial equations:

$$\begin{cases} w_i^j(\mu) - \nu_i^j(\mu)\mathbf{1} - \lambda_i^j(\mu) = 0, & \lambda_i^j(\mu) \geq 0, \\ g_i^j(\mu) \geq 0, & h_i^j(\mu) = 0, & g_{i,a}^j(\mu)\,\lambda_{i,a}^j(\mu) = 0, \end{cases} \quad \forall i \in [\![n]\!], \ j \in [\![\ell_i]\!], \ a \in [\![m_i^j]\!]. \quad \text{(KKT)}$$

Summarizing, in the multi-player case, and following Nie & Tang (2024), a behavioral NE can be computed by applying the Moment-SOS hierarchy to the semi-algebraic set defined by the collection of all coupled KKT conditions. We refer to the resulting hierarchy as the *KKT-based hierarchy*.

An important feature of this hierarchy is that it exhibits *finite convergence*, in contrast to the standard asymptotic convergence guarantee. This is achieved under canonical *genericity* assumptions: a property is called *generic* if it holds for all inputs outside a Lebesgue-measure-zero set in the coefficient space. For a fixed (multi-)degree, a polynomial is generic if its coefficient vector is generic. Nie & Tang (2024, Theorem A.1) show that for a polynomial game with generic utilities and constraints, the KKT equations admit only finitely many complex solutions. Consequently, the corresponding complex variety is finite, and existing results (Laurent, 2008; Lasserre et al., 2008) imply finite convergence of the hierarchy. However, applying this result directly in our setting requires care, since our constraints are not generic but correspond to simplices. Nevertheless, as shown earlier the constraints satisfy LICQ, which turns out to be the only property required in the proof of Nie & Tang (2024, Theorem A.1) (i.e., genericity is used there only to imply LICQ).

**Vertex-restricted Moment-SOS hierarchy.** An important structural feature of IREFGs is *absentmindedness*. Absentmindedness refers to the situation where a player cannot distinguish between two nodes that lie along the same history. Formally, a player is absentminded if there exist nodes $\pi_1, \pi_2 \in I$ with $\pi_1 \neq \pi_2$ such that $\pi_1$ lies on the unique path from $r$ to $\pi_2$—that is, multiple nodes along the same history belong to the same information set (cf. Figure 2.1 (b)). Whether a player is absentminded has important implications for the corresponding polynomial utility.

**Proposition 3.2.** *In non-absentminded IREFGs (NAM-IREFGs), each player's utility $u_i(\mu)$ is multi-affine in the blocks $\{\mu_i^j = (\mu_{i,a}^j)_{a=1}^{m^j}\}_{j=1}^{\ell_i}$, i.e., for any player i and infoset j, the map $\mu_i^j \mapsto u_i(\mu)$ is affine when all other blocks $\{\mu_{i'}^{j'}\}_{(i',j') \neq (i,j)}$ are held fixed.*

The proof is given in Appendix F.1. Proposition 3.2 has an interesting consequence for the Moment-SOS hierarchy in NAM-IREFG case. Indeed, since $u_i$ is affine in each block $\mu_i^j$, and the feasible region is a product of simplices, every blockwise maximization attains its value at an extreme point. More precisely, fixing all variables except $\mu_i^j$ (denote this by $\mu^{-(i,j)}$), the map $\mu_i^j \mapsto u_i(\mu)$ is affine, hence its maximum over $\Delta^{m_i^j}$ is achieved at a vertex $e_{i,a}^j$. Starting from any feasible profile, successively replacing each block $\mu_i^j$ by a maximizing vertex with respect to the current $\mu^{-(i,j)}$ never decreases the objective. After finitely many replacements we obtain a global maximizer whose blocks are all vertices. Defining $b_{i,a}^j(\mu) := \mu_{i,a}^j(\mu_{i,a}^j - 1)$, we see that instead of working with the product of simplices, we can equivalently run the hierarchy over the semi-algebraic vertex-restricted set

$$\mathcal{S}_{i,\text{vr}} := \big\{ \mu_i : h_i^j(\mu) = 0\ \forall j,\ \ b_{i,a}^j(\mu) = 0\ \forall j,a \big\}, \tag{6}$$

since $\max_{\mu_i \in \mathcal{S}_i} u_i(\mu) = \max_{\mu_i \in \mathcal{S}_{i,\text{vr}}} u_i(\mu)$. We refer to the resulting construction as the *vertex-restricted hierarchy*. In subsequent sections, we explore how this construction enables finite convergence guarantees in the single-player case, and yields computational savings in the multi-player case.

## 4 SINGLE-PLAYER IREFGS

In this section, we focus on *single-player* IREFGs. As explained in the previous section, this setting corresponds to a polynomial optimization problem over a product of simplices, to which the Moment-SOS hierarchy readily applies. We summarize our results for the single-player case below. In addition, Appendix C provides a summary of known complexity results, together with a new hardness result that leverages the connection to POPs.

**Theorem 4.1.** *Consider a single-player IREFG $\mathcal{G}$ with utility function $u$. Let $\ell$ be the number of infosets, $d_0 := \max_{j,a}\{\lceil \deg(u)/2 \rceil, \lceil \deg(g_a^j)/2 \rceil, \lceil \deg(h^j)/2 \rceil\}$, and $u^*$ be the ex-ante optimal value of $\mathcal{G}$. Denote by $u_d^{\text{sos}}, u_d^{\text{sos,kkt}}, u_d^{\text{sos,vr}}$ the values obtained from the SOS-Moment hierarchies applied respectively to the vanilla product-of-simplices, KKT-based, and vertex-restricted formulations. Similarly, we use the superscript $\text{mom}$ to denote the moment hierarchy. Then we have the following:*

*(i)* $\lim_{d \to \infty} u_d^{\text{sos}} = \lim_{d \to \infty} u_d^{\text{mom}} = u^*$.

*(ii) If $u$ is generic, there exists $d \geq d_0$ with $u_d^{\text{mom,kkt}} = u_d^{\text{sos,kkt}} = u^*$.*

*(iii) If $\mathcal{G}$ is non-absentminded, the degree-$(\ell+1)$ moment relaxation of the vertex-restricted problem is exact: $u_{\ell+1}^{\text{mom,vr}} = u^*$.*

*Proof sketch.* (i) follows from the standard convergence result for Lasserre's hierarchy; see, e.g., Putinar (1993); Lasserre (2001); Laurent (2009); Lasserre (2024). (ii) uses that, if $u$ is generic, the KKT system has finitely many solutions, so the feasible set of the augmented problem is a finite real variety. On such finite varieties the Lasserre hierarchy is finitely exact (Laurent, 2008; Lasserre et al., 2008). (iii) relies on the key observation that the ranks of the moment matrices stabilize once the degree exceeds the number of infosets $\ell$ (cf. Lemma F.1); by the flat extension theorem (Curto & Fialkow, 2000), this rank stabilization at degree $\ell + 1$ implies exactness at that level.

This result capitilizes on the connection between polynomial optimization and IREFGs. While convergence is asymptotic in general, for *almost all* single-player IREFGs, convergence is instead finite. Moreover, in the special case of NAM-IREFGs, we obtain finite convergence at an *explicit level* of the hierarchy which depends on the number of infosets in the game. The full version of the proof for this result is deferred to Appendix F.2.

Crucially, we can also *extract* ex-ante optima from the Moment-SOS hierarchies (the full procedure is given in Appendix D.4). In the NAM setting, flatness occurs at a fixed order determined by the number of infosets (**??**), so extraction is guaranteed at level $\ell + 1$. Under genericity, the KKT-based formulation allows one to extract a solution once finite convergence occurs.

# 5 MULTI-PLAYER IREFGS

In this section, we turn our attention to the computation of behavioral NE in *multi-player* IREFGs. Unlike the single-player case, NE are not guaranteed to exist (Wichardt, 2008), and indeed even deciding if one exists is hard (Tewolde et al., 2024). We therefore adopt the *select-verify-cut (SVC)* framework of Nie & Tang (2024), which searches for an NE (or certifies nonexistence) by solving a sequence of polynomial subproblems with Moment-SOS relaxations. Specifically, the SVC method searches for an NE by iterating over three steps:

(i) *Select:* Solve a selector program over the joint (KKT) system (plus any accumulated cuts) to pick an NE candidate.

(ii) *Verify:* For each player, fix opponent behavior at the candidate NE and solve a unilateral best-response problem. If no player can improve, the candidate is an NE; otherwise, extract a violated valid inequality from a deviator's best response.

(iii) *Cut:* Add new valid inequalities to the selector and re-solve. Each cut removes the non-NE candidate while preserving all NEs.

The method is iterated until a candidate passes verification (NE found) or the selector becomes infeasible (nonexistence certified). A full description of the method is given in Appendix E. Every subproblem in the SVC loop is a polynomial optimization problem, so we can solve them using the Moment-SOS hierarchy. See also Nie & Tang (2024, Section 4) for more details.

In general, SVC needs to iterate: each round solves one selector Moment-SOS relaxation and up to $n$ verification relaxations (one per player). By contrast, in NAM-IREFGs we can dispense with the verify/cut phases and solve a *single* vertex-restricted Moment-SOS relaxation to compute an NE or certify nonexistence, yielding substantial computational savings. Indeed, since $u_i(\cdot, \mu_{-i})$ is linear in $\mu_i$ (Proposition 3.2), the "no profitable deviation by player $i$" NE condition is equivalent to the family of vertex inequalities $u_i(\mu_i, \mu_{-i}) \geq u_i(v_i, \mu_{-i})$ for all $v_i \in \mathcal{S}_{i,\mathrm{vr}}$. Embedding these inequalities directly into the *Select* step gives the one-shot vertex-restricted program:

$$\min_{\mu} \ \varphi_\Theta(\mu) := [\mu]_1^\top \Theta [\mu]_1 \quad \text{s.t.} \quad \begin{cases} \mu \text{ satisfies (KKT)}, \\ u_i(\mu_i, \mu_{-i}) - u_i(v_i, \mu_{-i}) \geq 0, \ \forall v_i \in \mathcal{S}_{i,\mathrm{vr}}, \ \forall i, \end{cases} \tag{7}$$

where $[\mu]_1 = (1, \mu^\top)^\top$ and $\Theta \succ 0$ is generic. Feasibility of Equation (7) is equivalent to the existence of a behavioral NE; any feasible point is an NE, and the objective $\varphi_\Theta$ merely selects one among multiple solutions (if any). For instance, replacing $\varphi_\Theta$ with $\max_\mu \sum_{i=1}^n u_i(\mu)$ under the same constraints yields a *welfare-maximizing* NE. Therefore, in the NAM setting, the verify and cut phases are unnecessary.

**Theorem 5.1.** *Let $\mathscr{G}$ be a multi-player IREFG with utility functions $u_i$ for each player $i$. Throughout, subproblems are solved by the KKT-based hierarchies of increasing order. Then, we have the following:*

*(i) The SVC procedure is asymptotically exact: as the relaxation order and number of iterations grows, it returns a behavioral NE when one exists, and otherwise a certificate of nonexistence.*

*(ii) If $u_i$ are all generic, the KKT-based hierarchy has finite convergence for all SVC subproblems, and the SVC loop terminates in finitely many iterations.*

*(iii) If $\mathscr{G}$ is non-absentminded, the Verify/Cut phases in SVC are unnecessary: a single vertex-restricted Select (Equation (7)) suffices to compute an NE or certify nonexistence. Its Moment-SOS hierarchy is asymptotically exact; if $u_i$ are generic, it attains exactness at a finite order.*

*Proof sketch.* (i) combines the standard asymptotic convergence of Lasserre's hierarchy with the SVC steps: each loop either eliminates a non-NE candidate via a violated deviation inequality or certifies NE (or non-existence via selector infeasibility) (Nie & Tang, 2024). (ii) then follows from the fact that, under generic utilities, the KKT system has finitely many solutions, so only finitely many candidates can appear in SVC, and on such finite KKT sets the KKT-based SOS hierarchy is known to converge in finite order. (iii) uses the fact that in the non-absentminded case, "no profitable deviation" is equivalent to "no profitable vertex deviation", so a single vertex-restricted Select already

encodes the NE conditions. Its Moment–SOS hierarchy is asymptotically exact, and under generic utilities, finite exact.

The full version of proof for the above result is deferred to Appendix F.3. In analogy to the single-player case, Theorem 5.1 shows that finite convergence guarantees can be obtained for *almost all* IREFGs. NAM-IREFGs enjoy further computational improvements, needing only a single Moment-SOS hierarchy per iteration. Unlike the single-player case, explicit convergence at a fixed game-dependent level is not guaranteed, further highlighting the challenging nature of multi-player IREFGs.

## 6  (SOS)-Concave and (SOS)-Monotone IREFGs

In this section, we focus on defining tractable subclasses of IREFGs, and show how our methods obtain improved convergence guarantees in these subclasses. In the study of continuous games, the seminal work of Rosen (1965) introduced concave and monotone games, which exhibit desirable properties—in concave games, NE always exist, and in strictly monotone games, a unique NE exists. Leveraging the connection between IREFGs and polynomials, the following definitions are immediate:

An IREFG $\mathscr{G}$ is *concave* if, for every $i \in \mathcal{N}$, the map $\mu_i \mapsto u_i(\mu_i, \mu_{-i})$ is concave on $\mathcal{S}_i$ for every fixed $\mu_{-i} \in \mathcal{S}_{-i}$. Since $\mathscr{G}$ is polynomial, this is equivalent to the block Hessians being negative semidefinite: $\mathbf{H}_{u_i}(\mu) := \nabla^2_{\mu_i} u_i(\mu) \preceq 0, \ \forall \mu \in \mathcal{S}, \ \forall i \in \mathcal{N}$.

Following Rosen (1965), we collect the block partial derivatives into the pseudo-gradient: $v(\mu) := \left(\nabla^\top_{\mu_1} u_1(\mu), \ldots, \nabla^\top_{\mu_n} u_n(\mu)\right)^\top$. Let $\mathbf{J}(\mu) := \nabla v(\mu)$ denote its Jacobian, and define the symmetrized Jacobian $\mathbf{SJ}(\mu) := \frac{1}{2}\left(\mathbf{J}(\mu) + \mathbf{J}(\mu)^\top\right)$. Then, an IREFG $\mathscr{G}$ is *monotone* if $\langle v(\mu) - v(\nu), \ \mu - \nu \rangle \leq 0, \ \forall \mu, \nu \in \mathcal{S}$. Since $\mathscr{G}$ is polynomial, monotonicity holds if and only if the symmetrized Jacobian of $v$ is negative semidefinite (Rockafellar & Wets, 1998): $\mathbf{SJ}(\mu) \preceq 0, \ \forall \mu \in \mathcal{S}$.

While monotonicity immediately implies concavity, the converse does not hold. Moreover, we can define strict notions of concavity and monotonicity where the block Hessians and symmetrized Jacobian of a game are negative definite, respectively. With these notions in place, we recall the classical existence/uniqueness guarantees (Rosen, 1965, Theorems 1 and 2) tailored to our setting.

**Proposition 6.1.** *Let $\mathcal{S}$ be nonempty, convex, and compact, and let each $u_i$ be continuous.*

*(i) If the IREFG $\mathscr{G}$ is concave, then a behavioral Nash equilibrium exists.*

*(ii) If the IREFG $\mathscr{G}$ is strictly monotone, then the behavioral Nash equilibrium is unique.*

With respect to the SVC method utilized in Section 5, when an IREFG is concave, a joint profile $\mu$ is a Nash equilibrium *if and only if* it is a (KKT) point. Since the strategy set $\mathcal{S}$ is nonempty, convex, and compact, Proposition 6.1 guarantees existence, so the *Select* step is always feasible and returns an NE without any cuts.

**Corollary 6.2.** *If the IREFG $\mathscr{G}$ is concave, then any feasible point of the Select step (without any cuts) is an NE. Furthermore, if $\mathscr{G}$ is strictly monotone, the NE is unique and the Select step returns that unique equilibrium.*

While concavity and monotonicity are widely studied, it is in general hard to *verify* these properties (Ahmadi et al., 2013; Leon et al., 2025). Towards improving the tractability of these classes, Helton & Nie (2010) introduced 'effective' SOS-variants thereof, which are verifiable using a single SDP. In particular, we call an IREFG $\mathscr{G}$ *SOS-concave* if, for every $i \in \mathcal{N}$, the negative block Hessian is an SOS-matrix polynomial on $\mathcal{S}$, i.e., there exists a real matrix polynomial $F_i(\mu)$ such that $-\mathbf{H}_{u_i}(\mu) = F_i(\mu)F_i(\mu)^\top, \ \forall \mu \in \mathcal{S}, \ \forall i \in \mathcal{N}$. Furthermore, we call $\mathscr{G}$ *SOS-monotone* if its negative symmetrized Jacobian is an SOS-matrix polynomial, i.e., there exists a real matrix polynomial $F(\mu)$ such that $-\mathbf{SJ}(\mu) = F(\mu)F(\mu)^\top, \ \forall \mu \in \mathcal{S}$.

These are subclasses of concave and monotone IREFGs respectively, so SOS-concavity implies concavity and SOS-monotonicity implies monotonicity. However, the converse does not hold since there exist polynomials which are convex but not SOS-convex (Ahmadi & Parrilo, 2012). Going forward, we explore how these notions can be used to further improve the convergence

guarantees of our methods in single-player IREFGs. We note that the definitions of concavity and monotonicity coincide in the single-player case (cf. Proposition F.2). We show that for strict and SOS-concave/monotone single-player IREFGs, our proposed methods obtain stronger convergence guarantees:

**Theorem 6.3.** *Consider a single-player IREFG $\mathscr{G}$ with utility function $u$. Let $d_0 := \max_{j,a}\{\lceil \deg(u)/2 \rceil, \lceil \deg(g_a^j)/2 \rceil, \lceil \deg(h^j)/2 \rceil\}$. Then, the following holds:*

*(i) If $\mathscr{G}$ is strictly concave/monotone, then the Moment-SOS hierarchy has finite convergence: there exists $d \geq d_0$ such that $u_d^{\mathrm{sos}} = u_d^{\mathrm{mom}} = u^*$.*

*(ii) If $\mathscr{G}$ is SOS-concave/SOS-monotone, the degree-$d_0$ Moment-SOS relaxations are exact: $u_{d_0}^{\mathrm{mom}} = u_{d_0}^{\mathrm{sos}} = u^*$, i.e., the Moment-SOS hierarchy converges at the first level.*

The proof follows the same overall line of argument as (Lasserre, 2009a, Theorems 3.3 and 3.4), but adapted to our IREFG setting; full details are given in Appendix F.4. In the case of strictly concave/monotone single-player IREFGs, we obtain a sharper, finite-time guarantee of convergence to the unique behavioral NE without needing to modify the vanilla Moment-SOS hierarchy. Moreover, in SOS-concave/SOS-monotone single-player IREFGs, we show that convergence is possible at the first level of the hierarchy, requiring only a single SDP. Furthermore, by leveraging known results (Laurent, 2008; Lasserre, 2009a), once relaxation is exact, optimal solutions can be obtained directly from the first moments. In particular, for any optimal moment vector $y^*$, $\mu^* = (y_{e_a^j}^*)_{j,a}$. Thus, the extraction procedure as outlined in Appendix D.4 is no longer necessary, enabling efficient extraction of ex-ante optima.

## 7 Discussion and Future Work

In this paper, we applied Moment-SOS hierarchies to compute ex-ante optima in single-player IREFGs, and Nash equilibria in multi-player IREFGs. We showed that almost all IREFGs enjoy finite convergence. Moreover, in the case of NAM-IREFGs, the convergence is finite and dependent on the number of infosets in the game. We also propose tractable subclasses of IREFGs, and efficiently certifiable SOS-variants thereof. This leads to a subclass of SOS-concave/SOS-monotone single-player IREFGs which can be solved with a single SDP. Our results constitute an initial step towards better understanding the tractability of equilibrium computation in IREFGs, bringing together ideas from both extensive-form game theory and polynomial optimization.

Future work includes finding and analyzing other tractable subclasses of IREFGs, and designing faster algorithms that incorporate decentralized methods to compute KKT points. Moreover, while Statement (i) of Theorem 6.3 holds for the Moment-SOS approach, first-order decentralized methods are known to efficiently solve strictly monotone games (see e.g. Cai & Zheng (2023); Ba et al. (2025) and references therein). A natural question is how these approaches compare, both theoretically and empirically. Finally, while we have implemented our methods on some simple examples in Appendix G, implementing more tractable SDP solvers to improve scalability (Ahmadi & Majumdar, 2019; Monteiro et al., 2024; Han et al., 2025) remains a crucial direction to explore.

### Ethics Statement

While our results are primarily theoretical, we acknowledge that there could be potential societal and ethical consequences of our work, none of which we feel must be specifically highlighted.

### Reproducibility Statement

The code used to generate the examples in our paper are given in a supplementary file, and use standard Julia scientific computing packages, alongside the SumOfSquares package (Legat et al., 2017; Weisser et al., 2019).

### Large Language Model Usage Statement

LLMs were used sparingly to polish writing and grammar. They were not used in the ideation and analysis of our main theoretical results.

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

## A  ADDITIONAL RELATED WORK

**Sum-of-Squares in Polynomial Games.**   Outside of the works cited in the introduction, few other works have studied IREFGs from the perspective of polynomial optimization. However, games with polynomial utility functions have been proposed and studied in the past Dresher et al. (1950). Parrilo (2006); Laraki & Lasserre (2012) used semidefinite programming methods to find the value of two-player zero-sum polynomial games, and Stein et al. (2008) applied similar techniques to separable games (where utilities take a sum-of-products form). Recently, Nie & Tang (2023; 2024) also used semidefinite programming techniques to solve for Nash equilibria in $n$-player polynomial games under certain genericity assumptions, or otherwise detect the nonexistence of equilibria. Moreover, Bach (2025) studied sum-of-squares relaxations for min-max problems, deriving convergence results of their proposed hierarchies. Finally, Leon et al. (2025) used sum-of-squares hierarchies to certify concavity and monotonicity in polynomial games.

**Equilibrium Computation in IREFGs.**   Despite the hardness of computing equilibria in IREFGs, some work has studied the special case of two-player, zero-sum IREFGs and derived approximate algorithms to solve them. In particular, Bosansky et al. (2016); Čermák et al. (2017b) used MILP techniques to solve for minmax-optimal strategies. In the case of non-absentminded two-player zero-sum IREFGs, Čermák et al. (2018) also used MILP-based methods to obtain scalable algorithms that can approximate minmax strategies. In the case where the IREFGs are timeable and two-player zero-sum (a subclass of non-absentminded two-player zero-sum games), Zhang & Sandholm (2022); Zhang et al. (2023) utilized tree-decomposition based methods to obtain LP/CFR bounds for computing team-correlated (mixed) equilibria.

## B  FROM IREFGs TO POLYNOMIALS AND BACK

The equivalence between IREFGs and polynomial optimization is crucial to our proposed methods. In particular, the translation from IREFGs to polynomials is classical (Piccione & Rubinstein, 1997), and we give a full description here for clarity. First, let $P(h'|\mu, h)$ denote the realization probability of reaching $h'$ given that players using strategy $\mu$ are at state $h$. Note that if $h \notin \text{hist}(h')$ (i.e., if $h'$ is not reachable from $h$) then the probability is $0$. Intuitively, the realization probability given a behavioral strategy is just the product of choice probabilities along the path from $h$ to $h'$. In order to formally define $P(h'|\mu, h)$, we will need some additional notation. First, any node $h \in \mathcal{H}$ uniquely corresponds to a history $\text{hist}(h)$ from root $r$ to $h$.

- Function $\delta(h) : \mathcal{H} \to \mathbb{N}$ denotes the depth of the game tree starting from node $h \in \mathcal{H}$.
- Function $\nu(h, d) : \mathcal{H} \times \mathbb{N} \to \mathcal{H}$ identifies the node ancestor at depth $d \leq \delta$ from node $h$.
- Function $\alpha(h, d) : \mathcal{H} \times \mathbb{N} \to \cup_{h \in \mathcal{H}} \mathcal{A}(h)$ identifies the action ancestor at depth $d \leq \delta$ from node $h$.

Together, the sequence $(\nu(h, 0), \nu(h, 1), \ldots, \nu(h, \delta(h)))$ uniquely identifies the history of nodes from $r$ to $h$. Likewise, the sequence $(\alpha(h, 0), \alpha(h, 1), \ldots, \alpha(h, \delta(h) - 1))$ uniquely identifies the history of actions taken from $r$ to $h$. Then, the realization probability of node $h'$ from $h$ if the players use joint strategy profile $\sigma$ is given by:

**Definition B.1** (Realization Probability)**.**

$$P(h'|\mu, h) = \prod_{j=\delta(h')}^{\delta(h)-1} \mu(\alpha(h', j)|I_{\nu(h', j)}) \quad \text{if } h \in \text{hist}(h').$$

**Definition B.2** (Expected Utility for Player $i$)**.** *For player $i$ at node $h \in \mathcal{H} \setminus \mathcal{Z}$, if strategy profile $\mu$ is played, their expected utility is given by $u_i(\mu|h) := \sum_{z \in \mathcal{Z}} (P(z|\mu, h) \cdot \rho_i(z))$. In its complete form, we can write the expected utility for each player as follows:*

$$u_i(\mu) = \sum_{z \in \mathcal{Z}} \left( \prod_{j=0}^{\delta(z)-1} \mu(\alpha(z, j)|I_{\nu(z, j)}) \cdot \rho_i(z) \right)$$

With some abuse of notation, we can write $P(h|\mu) := P(h|\mu, r)$ where $r$ is the root node, and similarly $u_i(\mu) := u_i(\mu|r)$. Notice that by definition, the expected utility of each player is a polynomial function. In particular, $P(z|\mu, h) \cdot \rho_i(z)$ is a monomial in $\mu$ multiplied by a scalar.

Tewolde et al. (2023) also recently established that any polynomial can be transformed into a single-player IREFG. Subsequently, Tewolde et al. (2024) extended this result to a set of polynomials, and multi-player IREFGs. We report the single-player variant of the theorem below, and provide a concrete example to aid readability.

**Theorem B.3.** *Given a polynomial $p : \bigtimes_{j=1}^{\ell} \mathbb{R}^{m^j} \to \mathbb{R}$, we can construct a single-player extensive-form game with imperfect recall $\mathscr{G}$ such that the expected utility function of $\mathscr{G}$ satisfies $u(\mu) = p(\mu)$, with $\mu \in \bigtimes_{j=1}^{\ell} \Delta^{m^j - 1}$. Moreover, the construction can be done in polynomial time.*

*Proof.* The proof follows the analysis of Tewolde et al. (2023), with minor modifications to notation. Let $d_p = \deg(p)$ and write

$$p(x) = \sum_{D \in \mathrm{MB}(d, \boldsymbol{m})} \lambda_D \prod_{j,a=1}^{\ell, m^j} (x_a^j)^{D_a^j},$$

where $\boldsymbol{m} := (m^j)_{j=1}^{\ell}$, $\lambda_D$ are rational, and

$$\mathrm{MB}(d_p, \boldsymbol{m}) := \left\{ D = (D_a^j)_{j,a} \in \bigtimes_{j=1}^{\ell} \mathbb{N}_0^{m^j} : \sum_{j=1}^{\ell} \sum_{a=1}^{m^j} D_a^j \leq d_p \right\}.$$

Let $\mathrm{supp}(p) := \{D \in \mathrm{MB}(d_p, \boldsymbol{m}) : \lambda_D \neq 0\}$ and $|D| := \sum_{j,a} D_a^j$. For each $D$, define the multiset $\mathrm{supp}(D)^{\mathrm{ms}}$ that contains $D_a^j$ copies of the pair $(j, a)$ when $D_a^j > 0$. Then $|\mathrm{supp}(D)^{\mathrm{ms}}| = |D|$. The input encoding of $p$ consists of $\ell$, $(m^j)_{j=1}^{\ell}$, and the rational coefficients $(\lambda_D)_{D \in \mathrm{supp}(p)}$.

Given such a polynomial function, we build a single-player extensive-form game $\mathscr{G}$ with imperfect recall whose information sets are $I^j$ ($j \in [\ell]$), each with action set $\mathcal{A}_{I^j} = \{\tau_1, \ldots, \tau_{m^j}\}$. The game $\mathscr{G}$ has a chance root and depth at most $d_p + 1$. The chance node $h_0$ has one outgoing edge to a node $h_D$ for each $D \in \mathrm{supp}(p)$, and chance selects each $h_D$ with probability $1/|\mathrm{supp}(p)|$.

Fix a deterministic ordering $\prec$ of the multiset $\mathrm{supp}(D)^{\mathrm{ms}}$ (e.g., lexicographic on pairs $(j, a)$, repeated $D_a^j$ times). Initialize the *current edge* as the chance edge from $h_0$ into $h_D$.

- If $D = \boldsymbol{0}$ (the zero multi-index), make $h_D$ terminal with payoff $u(h_D) = \lambda_{\boldsymbol{0}} \cdot |\mathrm{supp}(p)|$.

- Otherwise, for each next element $(j, a)$ of $\mathrm{supp}(D)^{\mathrm{ms}}$ (in order $\prec$), do:

    - Insert a nonterminal decision node $h$ on the current edge and assign $h$ to the information set $I^j$.
    - Create $m^j$ outgoing edges from $h$, one for each action in $\mathcal{A}_{I^j} = \{\tau_1, \ldots, \tau_{m^j}\}$.
    - For every edge labeled $\tau_{a'}$ with $a' \neq a$, attach a terminal node with utility 0.
    - Update the *current edge* to be the unique edge labeled $\tau_a$.

After all elements of $\mathrm{supp}(D)^{\mathrm{ms}}$ have been processed, terminate the current edge with a terminal node $z_D$ and set its utility to $u(z_D) = \lambda_D \cdot |\mathrm{supp}(p)|$. This procedure yields a subtree $T_D$ of depth $|\mathrm{supp}(D)^{\mathrm{ms}}| = \sum_{j,a} D_a^j = |D|$.

In this reduction, any point $x = (x_a^j)_{j,a} \in \bigtimes_{j=1}^{\ell} \Delta^{m^j - 1}$ induces a behavioral strategy $\mu$ in $\mathscr{G}$ by $\mu(a_j \mid I^j) = x_a^j$ for all $j, a$. Let $z_D$ denote the terminal node associated with monomial index $D$ (including $D = \boldsymbol{0}$). At the chance root, $\mathbb{P}(h_D \mid \mu) = 1/|\mathrm{supp}(p)|$ for each $D \in \mathrm{supp}(p)$. If $D = \boldsymbol{0}$, then $z_D$ is reached immediately, so $\mathbb{P}(z_D \mid \mu) = 1/|\mathrm{supp}(p)|$. If $D \neq \boldsymbol{0}$, the loop that builds $T_D$ creates exactly $|D|$ decision nodes along the designated branch. At each visit to information set $I^j$, the unique continuing edge is labeled $\tau_a$ and is chosen with probability $x_a^j$. Since $(j, a)$ appears $D_a^j$

times in $\mathrm{supp}(D)^{\mathrm{ms}}$, we obtain

$$\mathbb{P}(z_D \mid \mu) \;=\; \frac{1}{|\mathrm{supp}(p)|} \prod_{j,a=1}^{\ell,m^j} (x_a^j)^{D_a^j}.$$

All sibling edges terminate with utility $0$ and do not contribute.

Therefore the expected utility is

$$
\begin{aligned}
u(\mu) &= \sum_{D \in \mathrm{supp}(p)} \mathbb{P}(z_D \mid \mu) \cdot \rho(z_D) \\
&= \sum_{D \in \mathrm{supp}(p)} \Big( \frac{1}{|\mathrm{supp}(p)|} \prod_{j,a} (x_a^j)^{D_a^j} \Big) \cdot \lambda_D \cdot |\mathrm{supp}(p)| \\
&= \sum_{D \in \mathrm{supp}(p)} \lambda_D \cdot \prod_{j,a} (x_a^j)^{D_a^j} \\
&= p(x).
\end{aligned}
$$

This extends to $u(\mu) = p(\mu)$ for all $\mu \in \bigtimes_{j=1}^{\ell} \Delta^{m^j-1}$.

For each $D \in \mathrm{supp}(p)$, $T_D$ contributes a path of length $|D| \leq d_p$ where, at each depth, at most $\max_j m^j$ leaves are created. Hence the total number of nodes and edges is

$$O\Big( \sum_{D \in \mathrm{supp}(p)} \big( |D| + |D| \cdot \max_j m^j \big) \Big) \;\subseteq\; O\big( |\mathrm{supp}(p)| \cdot d_p \cdot \max_j m^j \big).$$

All payoffs are rationals of the form $\lambda_D \cdot |\mathrm{supp}(p)|$, and the chance probabilities are $1/|\mathrm{supp}(p)|$, so labels are computable with bit complexity polynomial in the input size. Thus the game $\mathscr{G}$ is produced in polynomial time in the Turing model. $\qquad\square$

Clearly, this construction is not unique. Choosing a different total order on the multiset $\mathrm{supp}(D)^{\mathrm{ms}}$ yields a (potentially) different game tree. All such variants are payoff-equivalent, since the reach probability of $z_D$ depends only on the multiplicities $(D_a^j)$, hence $u(\mu) = p(\mu)$ in every case. For concreteness we fix the lexicographic order. Moreover, we provide a concrete example constructing a (single-player) IREFG from a polynomial.

**Example B.4.** *In this example, we index by (infoset, action) using subscripts to avoid ambiguity: the first subscript denotes the infoset and the second denotes the action (e.g., $x_{12}$). Let $\ell = 2$ with $m_1 = m_2 = 2$, and write $x_{11}, x_{12}$ for $I_1$ and $x_{21}, x_{22}$ for $I_2$. Consider*

$$p(x) = 2 + 3x_{11}x_{21} - 5x_{12}x_{22} + 4x_{21}^2.$$

*Then $\deg(p) = 2$. Fix the variable order $x_{11} \prec x_{12} \prec x_{21} \prec x_{22}$ and use the induced lexicographic order on multi-indices $D = (D_{11}, D_{12}, D_{21}, D_{22}) \in \mathbb{N}_0^4$. For each $D$, order the multiset $\mathrm{supp}(D)^{\mathrm{ms}}$ by listing the pairs $(j, a)$ in lexicographic order with multiplicity. With this convention,*

$$\mathrm{supp}(p) = \big\{ D^{(0)} = \mathbf{0},\ D^{(1)} = (1,0,1,0),\ D^{(2)} = (0,1,0,1),\ D^{(3)} = (0,0,2,0) \big\}$$

*so $|\mathrm{supp}(p)| = 4$ and the lexicographic order is $D^{(0)} \prec D^{(1)} \prec D^{(2)} \prec D^{(3)}$.*

*Then, we construct the game tree as follows:*

**Root (chance).** *Create a chance node $h_0$ with four equiprobable edges ($1/4$ each) to $h_{D^{(t)}}$, $t = 0, 1, 2, 3$.*

**Subtree $T_{D^{(0)}}$ (constant term).** *Make $h_{D^{(0)}}$ terminal with payoff $\rho = \lambda_{D^{(0)}} \cdot |\mathrm{supp}(p)| = 2 \cdot 4 = 8$.*

**Subtree $T_{D^{(1)}}$ for $x_{11}x_{21}$.** *Here $\mathrm{supp}(D^{(1)})^{\mathrm{ms}} = \{(1,1), (2,1)\}$. Process in order: (1) insert a node $h_1 \in I_1$; create two edges labeled $x_{11}, x_{12}$; attach $0$ to the $x_{12}$ edge; move along $x_{11}$. (2) insert a node $h_2 \in I_2$; create two edges labeled $x_{21}, x_{22}$; attach $0$ to the $x_{22}$ edge; move along $x_{21}$. Terminate with $z_{D^{(1)}}$ and payoff $\rho(z_{D^{(1)}}) = \lambda_{D^{(1)}} \cdot 4 = 12$.*

**Subtree $T_{D^{(2)}}$ for $x_{22}$.** *Here* $\mathrm{supp}(D^{(2)})^{\mathrm{ms}} = \{(1,2),(2,2)\}$. *Process in order: (1) insert $h_3 \in I_1$; create $x_{11}, x_{12}$; attach $0$ to $x_{11}$; move along $x_{12}$. (2) insert $h_4 \in I_2$; create $x_{21}, x_{22}$; attach $0$ to $x_{21}$; move along $x_{22}$. Terminate with $z_{D^{(2)}}$ and payoff $\rho(z_{D^{(2)}}) = \lambda_{D^{(2)}} \cdot 4 = -20$.*

**Subtree $T_{D^{(3)}}$ for $x_{21}^2$.** *Here* $\mathrm{supp}(D^{(3)})^{\mathrm{ms}} = \{(2,1),(2,1)\}$ *(two copies). Process in order: (1) insert $h_5 \in I_2$; create $x_{21}, x_{22}$; attach $0$ to $x_{22}$; move along $x_{21}$. (2) insert $h_6 \in I_2$; again $x_{21}$ continues, $x_{22}$ gets $0$. Terminate with $z_{D^{(3)}}$ and payoff $\rho(z_{D^{(3)}}) = \lambda_{D^{(3)}} \cdot 4 = 16$.*

*The constructed game tree is shown in Figure B.1.*

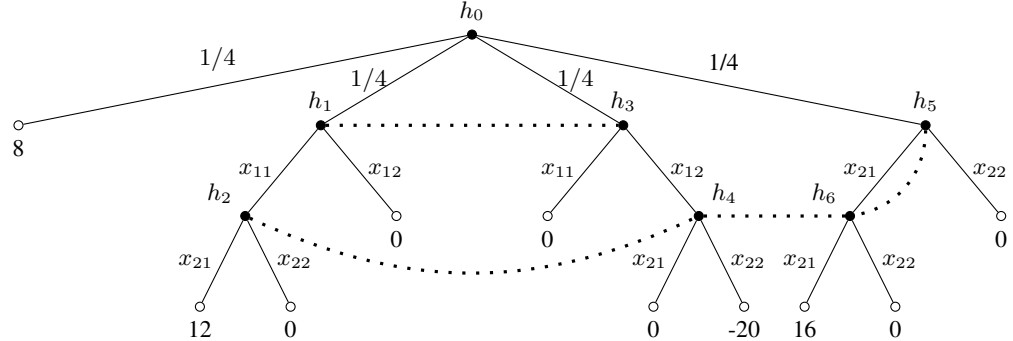

Figure B.1: Constructed Single-Player Imperfect-Recall Game for
$$p(x) = 2 + 3x_{11}x_{21} - 5x_{12}x_{22} + 4x_{21}^2.$$

**Verification.** *Given $x \in \Delta^1 \times \Delta^1$, define the behavioral strategy by $\mu(a_j \mid I^j) = x_a^j$. Then*

$$\mathbb{P}(z_{D^{(0)}} \mid \mu) = \tfrac{1}{4}, \quad \mathbb{P}(z_{D^{(1)}} \mid \mu) = \tfrac{1}{4}x_{11}x_{21}, \quad \mathbb{P}(z_{D^{(2)}} \mid \mu) = \tfrac{1}{4}x_{12}x_{22}, \quad \mathbb{P}(z_{D^{(3)}} \mid \mu) = \tfrac{1}{4}x_{21}^2.$$

*Hence*

$$
\begin{aligned}
u(\mu) &= \sum_{t=0}^{3} \mathbb{P}(z_{D^{(t)}} \mid \mu)\, \rho(z_{D^{(t)}}) \\
&= \tfrac{1}{4} \cdot 8 + \tfrac{1}{4}x_{11}x_{21} \cdot 12 + \tfrac{1}{4}x_{12}x_{22} \cdot (-20) + \tfrac{1}{4}x_{12}x_{21}^2 \cdot 16 \\
&= 2 + 3x_{11}x_{21} - 5x_{12}x_{22} + 4x_{21}^2 \\
&= p(x).
\end{aligned}
$$

## C  ON THE COMPUTATIONAL COMPLEXITY OF SINGLE-PLAYER IREFGS

While the computational hardness of the ex-ante problem (Equation (1)) has already been established in prior work (Tewolde et al., 2023), we note that one can extend recent results in the polynomial optimization literature to improve these complexity results. First, we introduce a hardness result for finding local optima over polytopes by Ahmadi & Zhang (2022):

**Lemma C.1** (Ahmadi & Zhang (2022),Theorem 2.6). *Unless $P = NP$, there is no polynomial-time algorithm that finds a point within Euclidean distance $c^n$ (for any constant $c \geq 0$) of a local minimizer of an $n$-variate quadratic function over a polytope.*

**Corollary C.2.** *Finding a local minimizer of a quadratic program over a simplex is NP-hard. Moreover, unless $P = NP$, there is no FPTAS for this problem.*

*Proof.* Specializing the argument in (Ahmadi & Zhang, 2022, Theorem 2.6, p. 7) by replacing $\lceil 3c^n \sqrt{n} \rceil$ by 1 yields the following: unless $P = NP$, there is no polynomial-time algorithm that finds an $\epsilon$-approximate local minimizer (for any constant $\epsilon \in [0, 0.5)$) of a quadratic program over a simplex. If an exact local minimizer could be computed in polynomial time, then, in particular, an $\epsilon$-approximate local minimizer (take $\epsilon = 0$) could also be computed in polynomial time, contradicting Lemma C.1. Therefore, computing an exact local minimizer over a simplex is NP-hard, and no FPTAS exists unless $P = NP$. $\square$

Hence, corresponding to single-player IREFGs, we have:

**Proposition C.3.** *Finding a local optimum of Equation* (1) *is NP-hard. Moreover, unless $P = NP$, there is no FPTAS for this problem. NP-hardness and conditional inapproximability hold even if the game instance $\mathcal{G}$ has no chance nodes, a tree depth of 2, and only one information set.*

Following this, we can improve the results from (Tewolde et al., 2023, Proposition 4):

**Proposition C.4.** *Finding the ex-ante optimal strategy $\mu^*$ of a given extensive-form game instance $\mathcal{G}$ is NP-hard. Specifically:*

1. *Unless $P = NP$, no FPTAS exists. NP-hardness and conditional inapproximability hold even if the game has no chance nodes, a tree depth of 2, and a single information set.*

2. *Unless $NP = ZPP$, no FPTAS exists. NP-hardness and conditional inapproximability hold even if the game has a tree depth of 3 and a single information set.*

3. *NP-hardness holds even without absentmindedness, with tree depth 4 and two actions per information set.*

4. *NP-hardness holds even without absentmindedness, with tree depth 3 and three actions per information set.*

*Proof.* To prove statement 1, we note that since any global optimum is in particular a local optimum, finding a global maximum must be at least as hard as finding a local one. Thus, the same hardness and inapproximability from Proposition C.3 immediately carry over to the problem of finding a global optimum, i.e., ex-ante optimal strategy $\mu^*$.

The statements 2-4 are proven in (Tewolde et al., 2023, Proposition 4). $\qquad\square$

Following Tewolde et al. (2023), we compile the core correspondences between the polynomial optimization formulation of Equation (1) and solution concepts in single-player IREFGs, together with the computational complexity of computing each notion, in Table C.1.

Table C.1: Correspondence between single-player IREFGs and POPs with complexity results.

| POP Optimality Notion | IREFG Equilibrium Notion | | Rel. | Complexity |
|---|---|---|---|---|
| Global maximizer | Ex-ante optimal strategy | | ⇔ | NP-hard |
| | (EDT,GDH) | with absentmindedness | ⇒ | |
| KKT point | | without absentmindedness | ⇔ | CLS-hard |
| | (CDT,GT) | | ⇔ | |

**Remark C.5.** *The three equilibrium notions for single-player IREFGs form an inclusion chain in general (Tewolde et al., 2023, Lemma 17): ex-ante optimal $\Rightarrow$ (EDT,GDH)-equilibrium $\Rightarrow$ (CDT,GT)-equilibrium. In games without absentmindedness, this strengthens to an equivalence (Tewolde et al., 2023, Lemma 13): a strategy is (CDT,GT)-equilibrium if and only if it is (EDT,GDH)-equilibrium. Moreover, when the game has a single information set, the blockwise argmax condition of (Tewolde et al., 2023, Lemma 15) reduces to a global argmax, so every (EDT,GDH)-equilibrium is also ex-ante optimal. Since ex-ante optimization is NP-hard, it follows that computing an (EDT,GDH)-equilibrium is already NP-hard in this one-infoset setting.*

# D THE MOMENT-SOS HIERARCHY

In this section, we present a thorough derivation of the Moment-SOS hierarchy which we have specialized to single-player IREFGs.

### D.1 PROBLEM REFORMULATION

Let $m = \sum_{j=1}^{\ell} m^j$. Denote the strategy variables collectively by $\mu = (\mu_a^j) \in \mathbb{R}^m$. The feasible set is

$$\mathcal{S} = \bigtimes_{j=1}^{\ell} \Delta(A_{I^j}) = \{\mu \in \mathbb{R}^m : \mu_a^j \geq 0 \ \forall j, a, \ \sum_{j=1}^{m_j} \mu_a^j = 1 \ \forall j\}. \tag{8}$$

Define constraint polynomials

$$g_a^j(\mu) = \mu_a^j, \quad h^j(\mu) = \sum_{j=1}^{m^j} \mu_a^j - 1. \tag{9}$$

By construction,

$$\mathcal{S} = \{\mu \in \mathbb{R}^m : g_a^j(\mu) \geq 0, j \in [\![\ell]\!], a \in [\![m^j]\!], \ h^j(\mu) = 0, j \in [\![\ell]\!]\}. \tag{10}$$

Accordingly, the quadratic module of $\mathcal{S}$ is

$$Q(\mathcal{S}) = \left\{ \sigma_0(\mu) + \sum_{j=1}^{\ell} \sum_{a=1}^{m^j} \sigma_a^j(\mu) \, g_a^j(\mu) + \sum_{j=1}^{\ell} p^j(\mu) \, h^j(\mu) : \sigma_0, \sigma_a^j \in \Sigma[\mu], \ p^j \in \mathbb{R}[\mu] \right\}. \tag{11}$$

An important property in our formulation is the Archimedean property, defined below:

**Definition D.1** (Archimedean property). *The quadratic module $Q(\mathcal{X})$ is* Archimedean *if there exists $N > 0$ such that*

$$N - \|x\|^2 \ \in \ Q(\mathcal{X}).$$

This property guarantees that $\mathcal{X}$ is compact.

For any $\mu \in \mathcal{S}$ one has $0 \leq \mu_a^j \leq 1$, hence $\|\mu\|^2 = \sum_{j,a} (\mu_a^j)^2 \leq \sum_{j,a} \mu_a^j = \sum_{j=1}^{\ell} \sum_{a=1}^{m^j} \mu_a^j = \ell$. Therefore the quadratic polynomial $g_0(\mu) := \ell - \|\mu\|^2$ is nonnegative on $\mathcal{S}$. Adding the redundant inequality $g_0(\mu) \geq 0$ to the description of $\mathcal{S}$ gives $\mathcal{S} = \{\mu : g_a^j(\mu) \geq 0, h^j(\mu) = 0, g_0(\mu) \geq 0\}$, so that $N - \|\mu\|^2 \in Q(\mathcal{S})$ with $N = \ell$. Hence, the quadratic module $Q(\mathcal{S})$ is Archimedean.

From Equation (1), the expected payoff of a single-player IREFG can be written as a polynomial $u(\mu)$. We seek

$$u^* = \max_{\mu \in \mathcal{S}} u(\mu). \tag{12}$$

### D.2 MOMENT-SOS RELAXATION

To approximate the nonconvex problem in Equation (12), we use Lasserre's Moment-SOS hierarchy. We derive a pair of dual hierarchies—one in the space of SOS multipliers (primal) and one in the space of moments (dual)—which yield provable upper bounds on $u^*$.

**(a) Primal (SOS-relaxation)**

Observe that

$$u^* = \inf\{t : t - u(\mu) \geq 0 \quad \forall \mu \in \mathcal{S}\} = \inf\{t : t - u(\mu) > 0 \quad \forall \mu \in \mathcal{S}\}. \tag{13}$$

Since $\mathcal{S} = \{\mu : g_a^j(\mu) \geq 0, \ h^j(\mu) = 0\}$ has Archimedean quadratic module $Q(\mathcal{S})$, Putinar's Positivstellensatz (Putinar, 1993) gives the equivalent infinite-dimensional certificate

$$u^* = \inf\{t : t - u(\mu) \in Q(\mathcal{S})\}. \tag{14}$$

However, membership in $Q(\mathcal{S})$ is still an infinite-dimensional constraint.

**Truncation to finite SDP.** Let $d_u := \lceil \deg(u)/2 \rceil$ and $d_{\mathcal{S}} := \max_{j,a}\{\lceil \deg(g_a^j)/2 \rceil, \lceil \deg(h^j)/2 \rceil\} = 1$, then $d_0 := \max\{d_u, d_{\mathcal{S}}\} = d_u$. Fix any relaxation order $d \geq d_0$. We truncate the sums-of-squares and polynomial multipliers to degree $2d$, obtaining:

$$u_d^{\text{sos}} = \inf_{t,\,\sigma_0,\{\sigma_a^j\},\{p^j\}} t \tag{15}$$
$$\text{s.t.} \quad t - u(\mu) \in Q_d(\mathcal{S}).$$

#### (b) Dual (Moment-relaxation)

We begin by expressing the original problem (12) in the space of Borel measures $\phi$ supported on $\mathcal{S}$. Since any admissible $\phi$ must satisfy $\phi \geq 0$ and $\int d\phi = 1$, one has the exact infinite-dimensional program

$$u^* = \sup_{\substack{\phi \in \mathcal{M}_+(\mathcal{S}) \\ \int d\phi = 1}} \int u(\mu)\, d\phi(\mu). \tag{16}$$

A measure $\phi$ is equivalently described by its full sequence of moments

$$y_\alpha = \int \mu^\alpha\, d\phi(\mu), \quad \forall \alpha \in \mathbb{R}^m. \tag{17}$$

Introduce the Riesz functional $L_y : \mathbb{R}[\mu] \to \mathbb{R}$ by $L_y(\mu^\alpha) = y_\alpha$, $\forall \alpha$. Expand $u(\mu) = \sum_\alpha u_\alpha \mu^\alpha$, so that

$$\int u(\mu)\, d\phi(\mu) = \sum_\alpha u_\alpha \int \mu^\alpha\, d\phi = \sum_\alpha u_\alpha y_\alpha =: L_y(u). \tag{18}$$

Requiring $\phi \geq 0$ and $\operatorname{supp}(\phi) \subseteq \mathcal{S}$ is equivalent to the following linear matrix constraints on $y$:

$$M_d(y) \succeq 0, \quad \forall d \in \mathbb{N} \quad \Longleftrightarrow \quad \int v(\mu)^2\, d\phi \geq 0, \quad \forall v \in \mathbb{R}[\mu], \tag{19}$$

$$M_d\big(g_a^j \star y\big) \succeq 0, \quad \forall d \in \mathbb{N} \quad \Longleftrightarrow \quad \int g_a^j(\mu)\, v(\mu)^2\, d\phi \geq 0, \quad \forall v \in \mathbb{R}[\mu], \tag{20}$$

$$L_y\big(h^j\, q\big) = 0, \quad \forall q \in \mathbb{R}[\mu], \quad \Longleftrightarrow \quad \int h^j(\mu)\, q(\mu)\, d\phi = 0, \quad \forall q \in \mathbb{R}[\mu], \tag{21}$$

$$y_0 = 1 \quad \Longleftrightarrow \quad \int 1\, d\phi = 1. \tag{22}$$

Thus Equation (16) can be rewritten as the (infinite-dimensional) moment program

$$u^* = \sup_y L_y(u) \tag{23}$$
$$\text{s.t.} \quad (19),\ (20),\ (21),\ (22).$$

**Truncation to finite SDP.** Fix any relaxation order $d \geq d_0$. In practice, we truncate Equation (23) to the degree-$d$ moment relaxation:

$$u_d^{\text{mom}} = \sup_y L_y(u)$$
$$\text{s.t.} \quad M_d(y) \succeq 0,$$
$$M_{d-1}\big(g_a^j \star y\big) \succeq 0, \quad \forall j, a, \tag{24}$$
$$L_y(h^j\, q) = 0, \quad \forall j,\ \forall q \in \mathbb{R}[\mu],\ \deg(h^j q) \leq 2d,$$
$$y_0 = 1.$$

### D.3 MOMENT-SOS HIERARCHY FOR NON-ABSENTMINDED GAMES

Recall that because $u$ is multi-affine, we have the following:

$$u^* = \max_{\mu \in \mathcal{S}} u(\mu) = \max_{\mu \in \mathcal{S}_{\mathrm{vr}}} u(\mu). \tag{25}$$

Let $d_0^{\mathrm{vr}} := \max\{d_u, d_{\mathcal{S}_{\mathrm{vr}}}\} = d_u$, and fix any relaxation order $d \geq d_0^{\mathrm{vr}}$. The degree-$d$ SOS relaxation of Equation (25) is

$$\begin{aligned}
u_d^{\mathrm{sos,vr}} = \inf \ & t \\
\text{s.t.} \quad & t - u(\mu) \in Q_d(\mathcal{S}_{\mathrm{vr}}).
\end{aligned} \tag{26}$$

The corresponding degree-$d$ moment relaxation reads

$$\begin{aligned}
u_d^{\mathrm{mom,vr}} = \sup_y \ & L_y(u) \\
\text{s.t.} \quad & M_d(y) \succeq 0, \\
& L_y(h^j\, q) = 0 \quad \forall j,\ \forall q \in \mathbb{R}[\mu],\ \deg(h^j q) \leq 2d, \\
& L_y(b_a^j\, q) = 0 \quad \forall j, a,\ \forall q \in \mathbb{R}[\mu],\ \deg(b_a^j q) \leq 2d, \\
& y_0 = 1.
\end{aligned} \tag{27}$$

Noth that there are no localizing PSD constraints for $\mu_a^j \geq 0$, because the binomials $b_a^j = 0$ already enforce $\mu_a^j \in \{0,1\} \subset [0,1]$.

### D.4 PSEUDO-EXPECTATIONS AND EXTRACTING SOLUTIONS

A feasible point $y = (y_\alpha)_{|\alpha| \leq 2d}$ of the truncated moment SDP in Equation (24) defines a linear functional $\widetilde{\mathrm{E}}_d : \mathbb{R}[\mu]_{2d} \to \mathbb{R}$ via $\widetilde{\mathrm{E}}_d[\mu^\alpha] = y_\alpha$. This functional behaves like an expectation operator up to degree $2d$: it is normalized ($\widetilde{\mathrm{E}}_d[1] = 1$), positive on squares ($\widetilde{\mathrm{E}}_d[q^2] \geq 0$ for all $q$ with $\deg q \leq d$), and it enforces feasibility through the linear identities induced by the constraints. Such a functional is often called a degree-$2d$ pseudo-expectation.

Expanding $u(\mu) = \sum_\alpha u_\alpha \mu^\alpha$, the moment objective is $L_y(u) = \sum_\alpha u_\alpha\, y_\alpha = \widetilde{\mathrm{E}}_d[u(\mu)]$. Hence the truncated moment relaxation in Equation (24) can be viewed as:

$$f_d^{\mathrm{mom}} = \sup\Big\{ \widetilde{\mathrm{E}}_d[u(\mu)] \ : \ \widetilde{\mathrm{E}}_d \text{ is a degree-}2d \text{ pseudo-expectation consistent with } (g \geq 0,\ h = 0) \Big\}.$$

In words, the primal SDP maximizes the pseudo-expected ex-ante payoff over all degree-$2d$ "virtual laws" that satisfy the polynomial feasibility conditions up to degree $2d$.

A feasible $y$ in Equation (24) need not come from any genuine probability measure on $\mathcal{S}$, and it generally encodes only a pseudo-expectation. A fundamental exception is the flat extension condition (Curto & Fialkow, 1996): if for some $s \leq d$,

$$\operatorname{rank} M_s(y) = \operatorname{rank} M_{s-1}(y), \tag{28}$$

then there exist atoms $\mu^{(1)}, \ldots, \mu^{(r)} \in \mathcal{S}$ with $r = \operatorname{rank}(M_s(y))$ and weights $\lambda_k \geq 0$ with $\sum_k \lambda_k = 1$ such that

$$\widetilde{\mathrm{E}}_s[p] = L_y(p) = \sum_{k=1}^r \lambda_k\, p(\mu^{(k)}) \qquad \forall\, p \in \mathbb{R}[\mu]_{2s}.$$

Thus a flat pseudo-expectation is the true expectation with respect to a finitely atomic probability measure supported on $\mathcal{S}$. Consequently, in the flat regime the SDP objective $\widetilde{\mathrm{E}}_s[u] = L_y(u)$ equals the true ex-ante payoff under optimal strategies, and the atoms $\{\mu^{(k)}\}$ (optimal solutions) can be extracted from $M_s(y)$ by standard linear-algebraic procedures (e.g., multiplication matrices).

### D.4.1 Extraction procedure

A standard sufficient flatness test certifying exactness and enabling solution extraction is $\operatorname{rank} M_s(y^*) = \operatorname{rank} M_{s-d_K}(y^*)$, where $d_K = d_{\mathcal{S}}$ in Equation (24), and $d_K = d_{\mathcal{S}_{\mathrm{vr}}}$ in Equation (27). Note that $d_{\mathcal{S}} = d_{\mathcal{S}_{\mathrm{vr}}} = 1$, then in both cases the flatness condition specializes to Equation (28).

Let $y^*$ be optimal for the degree-$d$ moment SDP and assume Equation (28) holds at some order $s \leq d$. Set $r := \operatorname{rank} M_s(y^*) = \operatorname{rank} M_{s-1}(y^*)$. Then $y^*$ admits an $r$-atomic representing measure $\sum_{k=1}^{r} \lambda_k \, \delta_{\mu^{(k)}}$ supported on the feasible set, with $\lambda_k > 0$ and $\sum_k \lambda_k = 1$. Since $L_{y^*}(u) = \sum_k \lambda_k u(\mu^{(k)})$ equals the global optimum, every atom satisfies $u(\mu^{(k)}) = u^*$ and hence is a global maximizer of $u$.

We now recover $\{\mu^{(k)}\}_{k=1}^{r}$ directly from the optimal moment matrix $M_s(y^*)$ by the standard multiplication-matrix routine. The extraction steps are based on Henrion & Lasserre (2005).

Let $v_s(\mu)$ be the vector of all monomials in $\mu$ of total degree $\leq s$, with length $N_s = \binom{m+s}{s}$. Then the order-$s$ moment matrix can be represented as:

$$M_s(y^*) \;=\; \sum_{k=1}^{r^*} \lambda_k \, v_s\big(\mu^{(k)}\big) \, v_s\big(\mu^{(k)}\big)^\top \;=\; V^*(V^*)^\top, \tag{29}$$

where $\lambda_k \geq 0$, $\sum_{k=1}^{r} \lambda_k = 1$, and $V^* \in \mathbb{R}^{N_s \times r}$ collects the columns $\sqrt{\lambda_k}\, v_s\big(\mu^{(k)}\big)$.

For computation, we form a rank factor of $M_s(y^*)$ by retaining the $r$ positive modes (e.g., via eigendecomposition or a Cholesky-type factorization):

$$M_s(y^*) \;=\; VV^\top, \qquad V \in \mathbb{R}^{N_s \times r}. \tag{30}$$

By construction, $\operatorname{span}(V) = \operatorname{span}(V^*)$. Hence the columns of $V$ are linear combinations of $\{\sqrt{\lambda_j}\, v_s(\mu^{(k)})\}_{k=1}^{r^*}$.

To obtain an explicit monomial basis of that subspace, we reduce $V$ to column-echelon form by Gaussian elimination with column pivoting, and rescale so that the pivot block is the identity. This gives $\widehat{V} = VT$, $\widehat{V}_{B,:} = I_{r^*}$, with $T \in \mathbb{R}^{r \times r}$ invertible and $B = \{\beta_1, \ldots, \beta_r\}$ the indices of the pivot rows (monomials). The pivot indices select a monomial "generating basis":

$$w(\mu) \;:=\; [\mu^{\beta_1} \; \cdots \; \mu^{\beta_r}]^T.$$

The same elimination step simultaneously produces linear reduction rules for all monomials of degree $\leq s$: each $\mu^\alpha$ is written, on the support, as a linear combination of the generators $\{\mu^{\beta_k}\}_{k=1}^{r}$. Stacking these relations row-wise yields the rewriting matrix $\mathcal{R} \in \mathbb{R}^{N_s \times r}$ of the form (identity in the pivot rows and coefficients elsewhere)

$$v_s(\mu) \;=\; \mathcal{R}\, w(\mu), \qquad \mathcal{R}_{B,:} = I_r, \tag{31}$$

which is exactly the coordinate change from the standard monomial vector $v_s$ to the generating basis $w$ on the atoms $\{\mu^{(k)}\}_{k=1}^{r}$.

In the basis $w$, multiplication by each coordinate $\mu_\eta$ acts linearly. For $\eta = 1, \ldots, m$ we build the multiplication matrices $N_\eta \in \mathbb{R}^{r \times r}$ defined by

$$N_\eta\, w(\mu) \;=\; \mu_\eta\, w(\mu). \tag{32}$$

Concretely, for the $k$th basis monomial $\mu^{\beta_k}$, form $\gamma = \beta_k + e_\eta$; if $\gamma \in B$ set $N_\eta(:,k) = e_{\mathrm{row}(\gamma)}$, otherwise take $N_\eta(:,k) = \mathcal{R}_{\gamma,:}^\top$ from Equation (31).

The atoms appear as common eigenpairs of $\{N_\eta\}$. Indeed, with $e_k := w(\mu^{(k)})$ one has $N_\eta e_k = \mu_\eta^{(k)} e_k$ for all $\eta$. For robust computation we form a random convex combination

$$N \;=\; \sum_{\eta=1}^{m} \lambda_\eta N_\eta, \qquad \lambda_\eta \geq 0, \; \sum_\eta \lambda_\eta = 1,$$

---

**Algorithm D.1:** Extraction for single-player IREFGs

---

**Input:** Optimal solution $y^*$; order $s$ with rank $M_s(y^*) = \text{rank}\, M_{s-1}(y^*) = r$.

**Output:** r optimal strategies $\{\mu^{(k)}\}_{k=1}^r$.

**1.** Factor $M_s = VV^\top$ as in Equation (30).

**2.** Build $\mathcal{R}$ so $v_s = \mathcal{R}w$ as in Equation (31).

**3.** For each $\mu_\eta$, build $N_\eta$ so $N_\eta\, w(\mu) = \mu_\eta\, w(\mu)$ as in Equation (32).

**4.** Form $N = \sum_\ell \lambda_\ell N_\ell$ with random $\lambda$; compute $N = QTQ^\top$ as in Equation (33).

**5.** For each $k$, set $\mu_\eta^{(k)} = q_k^\top N_\eta q_k$ for all $\eta$ as in Equation (34).

---

which generically has simple spectrum and shares the same eigenvectors. The ordered real Schur decomposition

$$N \;=\; QTQ^\top, \qquad Q = [q_1\; \cdots\; q_r] \tag{33}$$

returns orthonormal vectors $q_k$ spanning the eigenvectors $w(\mu^{(k)})$. The coordinates of each atom are then read by Rayleigh quotients:

$$\mu_\eta^{(k)} \;=\; q_k^\top N_\eta\, q_k, \qquad \eta = 1, \ldots, m, \;\; k = 1, \ldots, r. \tag{34}$$

The above extraction procedure is summarized in Algorithm D.1.

Since both $\mathcal{S}$ and $\mathcal{S}_{\mathrm{vr}}$ are nonempty, both Equation (24) and Equation (27) are always feasible (take, e.g., the Dirac measure at any $\mu \in \mathcal{S}$ or $\mathcal{S}_{\mathrm{vr}}$). We can thus solve them using the standard procedure:

**Moment-SOS loop.** Recall $d_0 = d_0^{\mathrm{vr}} = d_u$. Initialize $d := d_0$ and do:

1. Solve the order-$d$ moment SDP; obtain optimal $y_d$ and upper bound $u_d^{\mathrm{mom}} = L_{y_d}(u)$.

2. For $s = d_0, \ldots, d$, test the rank condition in Equation (28). If it holds for some $s$, *terminate*: the relaxation is exact and one can extract the global maximizers from $M_s(y_d)$ using Algorithm D.1.

3. Otherwise, increase the relaxation order: $d \leftarrow d + 1$ and go back to Step 1.

For single-player NAM-IREFGs, Lemma **??** shows that $\text{rank}\, M_s(y) = \text{rank}\, M_\ell(y)\ \forall s > \ell$, and Statement (iii) of Theorem 4.1 gives exactness at degree $\ell+1$: $u_{\ell+1}^{\mathrm{mom,vr}} = u_{\ell+1}^{\mathrm{sos,vr}} = u^*$. Hence the flatness test in Step 2 necessarily succeeds at $s = d = \ell+1$ (indeed for all $s > \ell$). The loop is therefore guaranteed to terminate at this fixed order, determined solely by the number of infosets, and the extraction returns at least one optimal pure strategy.

For single-player IREFGs with absentmindedness, the same loop produces a monotone sequence of upper bounds $u_d^{\mathrm{mom}} \downarrow u^*$ and terminates as soon as flatness is detected at some order $s$, which certifies exactness and enables extraction. In the absence of flatness, one increases $d$ to tighten the bound, with asymptotic convergence to $u^*$ guaranteed. If $u$ is generic, Statement (ii) of Theorem 4.1 ensures finite termination of the loop, with extraction of at least one certified global maximizer for the KKT-based problem.

# E  THE SELECT-VERIFY-CUT PROCEDURE

Recall the joint KKT system

$$\begin{cases} w_i^j(\mu) - \nu_i^j(\mu)\mathbf{1} - \lambda_i^j(\mu) = 0, & \lambda_i^j(\mu) \geq 0, \\ g_i^j(\mu) \geq 0, \quad h_i^j(\mu) = 0, \quad g_{i,a}^j(\mu)\, \lambda_{i,a}^j(\mu) = 0, & \end{cases} \quad \forall i \in [\![n]\!],\, j \in [\![\ell_i]\!],\, a \in [\![m_i^j]\!]. \tag{35}$$

The following exposition is based on the method introduced in Nie & Tang (2024).

**(i) Select.** Let $n_0 := \sum_{i=1}^{n} \sum_{j=1}^{\ell_i} m_i^j$ be the total dimension, set $[\mu]_1 = (1, \mu^\top)^\top$, and choose a generic positive definite matrix $\Theta \in \mathbb{R}^{(n_0+1) \times (n_0+1)}$. Then all NEs are feasible points of

$$\min_{\mu} \quad \varphi_\Theta(\mu) := [\mu]_1^\top \Theta [\mu]_1 \quad \text{s.t.} \quad \begin{cases} \mu \text{ satisfies (KKT)}, \\ u_i(\mu_i, \mu_{-i}) - u_i(v_i, \mu_{-i}) \geq 0, \quad \forall v_i \in K_i, \forall i, \end{cases} \tag{36}$$

where $K_i$ is the current (finite) set of deviation profiles used as cuts (initially $K_i = \emptyset$). If Equation (36) is infeasible, there is no NE. If it is feasible, a minimizer exists because the feasible set is compact and $\varphi_\Theta$ is continuous.

**(ii) Verify.** Let $\hat{\mu} \in \mathbb{R}^{n_0}$ be an optimizer of Equation (36). For each player $i$, evaluate the best-response improvement against $\hat{\mu}_{-i}$ by solving the KKT-restricted POP (same value as the unrestricted best-response since LICQ holds on products of simplices):

$$\omega_i := \max_{\mu_i} \quad u_i(\mu_i, \hat{\mu}_{-i}) - u_i(\hat{\mu}_i, \hat{\mu}_{-i})$$

$$\text{s.t.} \quad w_i^j(\mu) - \nu_i^j(\mu)\mathbf{1} - \lambda_i^j(\mu) = 0, \quad \lambda_i^j(\mu) \geq 0, \qquad \forall j, \tag{37}$$

$$g_i^j(\mu) \geq 0, \quad h_i^j(\mu) = 0, \quad g_{i,a}^j(\mu) \lambda_{i,a}^j(\mu) = 0, \qquad \forall j, a.$$

If every $\omega_i \leq 0$, no player can profitably deviate and $\hat{\mu}$ is an NE.

**(iii) Cut.** If some $\omega_i > 0$, take one or more maximizers $v_i \in \arg\max u_i(\mu_i, \hat{\mu}_{-i}) - u_i(\hat{\mu}_i, \hat{\mu}_{-i})$ and add the valid NE cuts

$$u_i(\mu_i, \mu_{-i}) - u_i(v_i, \mu_{-i}) \geq 0 \qquad (v_i \in K_i \leftarrow K_i \cup \{v_i\}), \tag{38}$$

which every NE satisfies but $\hat{\mu}$ violates; then resolve Equation (36) with the enlarged cut set. Each violated cut eliminates the current candidate while preserving the entire NE set. Repeat (select-verify-cut) until an NE is certified or nonexistence is proved.

# F OMITTED PROOFS FROM MAIN TEXT

## F.1 PROOFS FROM SECTION 3

**Proposition 3.2.** *In non-absentminded IREFGs (NAM-IREFGs), each player's utility $u_i(\mu)$ is multi-affine in the blocks $\{\mu_i^j = (\mu_{i,a}^j)_{a=1}^{m^j}\}_{j=1}^{\ell_i}$, i.e., for any player $i$ and infoset $j$, the map $\mu_i^j \mapsto u_i(\mu)$ is affine when all other blocks $\{\mu_{i'}^{j'}\}_{(i',j') \neq (i,j)}$ are held fixed.*

*Proof.* Fix a terminal history $z \in Z$. For each player $i$, let $\mathcal{I}_i(z) \subseteq [\ell_i]$ be the set of (distinct) infosets of player $i$ visited on the unique path to $z$. By non-absentmindedness, each $I_i^j$ is visited at most once. Let $a_i^j(z)$ be the action taken at $I_i^j \in \mathcal{I}_i(z)$, and let $c(z)$ denote the product of chance move probabilities (independent of $\mu$). The reach probability factorizes as

$$\mathbb{P}(z \mid \mu) = c(z) \prod_{i'=1}^{n} \prod_{j \in \mathcal{I}_{i'}(z)} \mu_{i', a_{i'}^j(z)}^j.$$

Hence player $i$'s expected payoff is

$$u_i(\mu) = \sum_{z \in Z} \rho_i(z) \mathbb{P}(z \mid \mu) = \sum_{z \in Z} \big(\rho_i(z)c(z)\big) \prod_{i'=1}^{n} \prod_{j \in \mathcal{I}_{i'}(z)} \mu_{i', a_{i'}^j(z)}^j.$$

In each summand, the dependence on the block $\mu_i^j$ is either absent (if $j \notin \mathcal{I}_i(z)$) or linear through a single coordinate $\mu_{i, a_i^j(z)}^j$ (if $j \in \mathcal{I}_i(z)$); by non-absentmindedness, no monomial contains two coordinates from the same block. Therefore, with all other blocks $\{\mu_{i'}^{j'}\}_{(i',j') \neq (i,j)}$ held fixed, the map $\mu_i^j \mapsto u_i(\mu)$ is affine on the simplex $\Delta^{m_i^j}$. Since this holds for every $(i,j)$, $u_i$ is multi-affine in the blocks $\{\mu_i^j\}_{j=1}^{\ell_i}$. $\qquad\square$

F.2 PROOFS FROM SECTION 4

**Theorem 4.1.** *Consider a single-player IREFG $\mathcal{G}$ with utility function $u$. Let $\ell$ be the number of infosets, $d_0 := \max_{j,a}\{\lceil \deg(u)/2 \rceil, \lceil \deg(g_a^j)/2 \rceil, \lceil \deg(h^j)/2 \rceil\}$, and $u^*$ be the ex-ante optimal value of $\mathcal{G}$. Denote by $u_d^{\text{sos}}, u_d^{\text{sos,kkt}}, u_d^{\text{sos,vr}}$ the values obtained from the SOS-Moment hierarchies applied respectively to the vanilla product-of-simplices, KKT-based , and vertex-restricted formulations. Similarly, we use the superscript* mom *to denote the moment hierarchy. Then we have the following:*

*(i)* $\lim_{d\to\infty} u_d^{\text{sos}} = \lim_{d\to\infty} u_d^{\text{mom}} = u^*$.

*(ii) If $u$ is generic, there exists $d \geq d_0$ with $u_d^{\text{mom,kkt}} = u_d^{\text{sos,kkt}} = u^*$.*

*(iii) If $\mathcal{G}$ is non-absentminded, the degree-$(\ell+1)$ moment relaxation of the vertex-restricted problem is exact: $u_{\ell+1}^{\text{mom,vr}} = u^*$.*

*Proof.* To prove Statement (i), note that because $Q(\mathcal{S})$ is Archimedean, the asymptotic convergence follows from Putinar's Positivstellensatz and Lasserre's hierarchy (Putinar, 1993; Lasserre, 2001; Laurent, 2009; Lasserre, 2024).

To prove Statement (ii), recall from Section 3 that if $u$ is generic (a property which holds for almost all single-player IREFGs), the KKT set is finite. Augmenting Equation (1) with the polynomial KKT system does not change the set of maximizers, but restricts feasibility to a finite real variety. For such finite varieties, the Lasserre hierarchy has finite convergence: for some $d$ large enough, $u_d^{\text{mom,kkt}} = u_d^{\text{sos,kkt}} = u^*$ and the flat extension (rank) condition holds, allowing recovery of $\mu^*$; this is immediate from e.g. Laurent (2008, Thm. 6.15) and Lasserre et al. (2008, Prop. 4.6).

In order to prove Statement (iii), we first establish a key rank-stabilization lemma:

**Lemma F.1.** *For every feasible solution $y$ of the degree-$s$ moment relaxation in Equation* (27) *with $s > \ell$, it holds that $\text{rank } M_s(y) = \text{rank } M_\ell(y)$.*

*Proof of Lemma F.1.* Let $v_s(\mu)$ collect all monomials of total degree $\leq s$ and recall $M_s(y) = L_y(v_s v_s^\top)$. Index the columns of $M_s(y)$ by monomials and write the block decomposition

$$M_s(y) = \begin{bmatrix} M_\ell(y) & B \\ B^\top & C \end{bmatrix},$$

where $M_\ell(y)$ is indexed by monomials of degree $\leq \ell$ and $B$ by monomials of degree $> \ell$.

Fix a column of $B$ indexed by $m(\mu) = \prod_{j=1}^\ell \prod_{a=1}^{m^j} (\mu_a^j)^{\alpha_a^j}$ with $\sum_{j,a} \alpha_a^j = \deg m > \ell$. Define the clipped monomial

$$\widehat{m}(\mu) := \prod_{j=1}^\ell \prod_{a=1}^{m^j} (\mu_a^j)^{\min\{1,\alpha_a^j\}},$$

so each exponent $\geq 1$ is replaced by 1.

By repeatedly using $L_y((\mu_a^j)^2 q) = L_y(\mu_a^j q)$ (i.e., $b_a^j = 0$), for any row index monomial $r$ (degree $\leq s$) we obtain
$$L_y(m(\mu)\,r(\mu)) = L_y(\widehat{m}(\mu)\,r(\mu)).$$
Hence the column of $M_s(y)$ indexed by $m$ coincides with the column indexed by $\widehat{m}$.

If $\widehat{m}$ uses at most one variable per block, then $\deg \widehat{m} \leq \ell$ and the column indexed by $m$ is *identical* to a column of $M_\ell(y)$. If, instead, $\widehat{m}$ contains two distinct variables from the same block (say $\mu_a^j$ and $\mu_{a'}^j$ with $a \neq a'$), then $\widehat{m}$ vanishes on the vertex set $\mathcal{S}_{\text{NAM}}$ (one-hot per block), so for all admissible rows $r$, $L_y(\widehat{m}\,r) = 0$, and the entire column indexed by $m$ is the zero vector.

Consequently, every column of $B$ is either zero or identical to a column of $M_\ell(y)$. Applying the same argument to the lower block, with $A := \begin{bmatrix} M_\ell(y) \\ B^\top \end{bmatrix}$ and $D := \begin{bmatrix} B \\ C \end{bmatrix}$, shows that every column of $D$ is either zero or identical to a column of $A$. Therefore the column space of $M_s(y)$ is contained

in the column space of $A$, hence $\operatorname{rank} M_s(y) \leq \operatorname{rank} A = \operatorname{rank} M_\ell(y)$. The reverse inequality is obvious because $M_\ell(y)$ is a principal submatrix of $M_s(y)$. Hence $\operatorname{rank} M_s(y) = \operatorname{rank} M_\ell(y)$ for all $s > \ell$. $\qquad\square$

Let $s := \ell+1$ and let $y^*$ be an optimal solution of the order-$s$ moment relaxation Equation (27). By Lemma F.1, $\operatorname{rank} M_s(y^*) = \operatorname{rank} M_\ell(y^*)$, so the flatness condition holds at order $s$.

Set $r := \operatorname{rank} M_s(y^*)$. By (Curto & Fialkow, 2000, Theorem 1.6), $y^*$ admits an $r$-atomic representing measure $\sum_{k=1}^r \lambda_j \, \delta_{\mu^{(k)}}$ supported on the feasible set, with $\lambda_k > 0$ and $\sum_k \lambda_k = 1$. Moreover, the equalities $L_{y^*}(h^j q) = L_{y^*}(b_a^j q) = 0$ in Equation (27) enforce $\operatorname{supp}(\mu^*) \subseteq \mathcal{S}_{\mathrm{vr}}$. Therefore,

$$L_{y^*}(u) \;=\; \sum_{k=1}^r \lambda_k \, u(\mu^{(k)}) \;\leq\; \max_{\mu \in \mathcal{S}_{\mathrm{vr}}} u(\mu) \;=\; u^*.$$

Because Equation (27) is a relaxation of the original problem, $u_s^{\mathrm{mom,vr}} \geq u^*$ for all $s$. At $s = \ell+1$, we have

$$u^* \;\leq\; u_{\ell+1}^{\mathrm{mom,vr}} \;=\; L_{y^*}(u) \;\leq\; u^*.$$

Hence $u_{\ell+1}^{\mathrm{mom,vr}} = u^*$ and the optimum is attained at $y^*$. $\qquad\square$

### F.3 Proofs from Section 5

**Theorem 5.1.** *Let $\mathscr{G}$ be a multi-player IREFG with utility functions $u_i$ for each player $i$. Throughout, subproblems are solved by the KKT-based hierarchies of increasing order. Then, we have the following:*

*(i) The SVC procedure is asymptotically exact: as the relaxation order and number of iterations grows, it returns a behavioral NE when one exists, and otherwise a certificate of nonexistence.*

*(ii) If $u_i$ are all generic, the KKT-based hierarchy has finite convergence for all SVC subproblems, and the SVC loop terminates in finitely many iterations.*

*(iii) If $\mathscr{G}$ is non-absentminded, the Verify/Cut phases in SVC are unnecessary: a single vertex-restricted Select (Equation (7)) suffices to compute an NE or certify nonexistence. Its Moment-SOS hierarchy is asymptotically exact; if $u_i$ are generic, it attains exactness at a finite order.*

*Proof.* (i) Let $\ell_0 := \sum_{i=1}^n \ell_i$ be the total number of infosets. For any feasible $\mu$ we have, for each infoset block, $\sum_{a=1}^{m_i^j} \mu_{i,a}^j = 1$ and $0 \leq \mu_{i,a}^j \leq 1$. Hence

$$\|\mu\|^2 = \sum_{i=1}^n \sum_{j=1}^{\ell_i} \sum_{a=1}^{m_i^j} (\mu_{i,a}^j)^2 \;\leq\; \sum_{i=1}^n \sum_{j=1}^{\ell_i} \sum_{a=1}^{m_i^j} \mu_{i,a}^j \;=\; \sum_{i=1}^n \sum_{j=1}^{\ell_i} 1 \;=\; \ell_0,$$

so $g_0(\mu) := \ell_0 - \|\mu\|^2 \geq 0$ on $\mathcal{S}$. Adding $g_0 \geq 0$ yields an Archimedean quadratic module, and the same holds for each verification feasible set. By standard results for Lasserre's hierarchy on Archimedean sets (see, e.g., Lasserre (2001); Laurent (2009)), every fixed selector (with a fixed cut set) and every verification problem is asymptotically exact: the moment optimal values converge to the true optima as $d \to \infty$, flat truncation recovers optimizers, and infeasibility is detected at high order.

At loop $t$, solve one selector and up to $n$ verifications. If the selector becomes infeasible at some order, nonexistence is certified and the procedure stops. Otherwise, let $\hat{\mu}^{(t)}$ be a selector optimizer recovered once flatness occurs. If all verification values are $\leq 0$, then $\hat{\mu}^{(t)}$ is an NE and we stop. If some player gains ($> 0$), extract one or more violated valid inequalities $u_i(\mu_i, \mu_{-i}) - u_i(v_i, \mu_{-i}) \geq 0$ from the deviator $v_i$ and add them to the selector. As relaxation orders increase across loops, subproblem solutions approach their true optima; any limit point of flat selector solutions satisfies all accumulated valid inequalities, i.e., the Nash conditions. Hence the method converges asymptotically to an NE, or certifies nonexistence if the selector turns infeasible.

(ii) Under generic utilities, which hold for almost all IREFGs, the joint KKT set is finite (cf. Nie & Tang (2024)). Then the selector's feasible set (joint KKT plus cuts) is finite. Each failed candidate is

removed by the new valid inequalities without excluding any NE, so only finitely many candidates can be visited before selecting an NE or proving infeasibility. On finite feasible sets, the Moment-SOS hierarchy attains finite convergence and yields atomic solutions (see, e.g., Laurent (2008); Lasserre et al. (2008)). Therefore both the select-verify-cut loop and its SDP subproblems terminate in finitely many steps.

(iii) In the NAM case, $u_i(\cdot, \mu_{-i})$ is linear in each block $\mu_i^j$ (Proposition 3.2), so "no profitable deviation by $i$" $\iff u_i(\mu_i, \mu_{-i}) \geq u_i(v_i, \mu_{-i}) \ \forall v_i \in \mathcal{S}_{i,\mathrm{rm}}$. Thus feasibility of the single vertex-restricted selector Equation (7) is equivalent to the existence of a behavioral NE, and the verify/cut phases are unnecessary.

To see asymptotic exactness, note that the feasible region of Equation (7) is contained in the product of simplices (per-player KKT equalities are imposed together with vertex deviation inequalities). Similar to (i), adding $g_0(\mu) = \ell_0 - \|\mu\|^2 \geq 0$ makes the quadratic module Archimedean. By standard results for Lasserre's hierarchy on Archimedean sets, the Moment-SOS relaxation of Equation (7) is asymptotically exact; flatness yields extraction, and infeasibility is detected at sufficiently high order.

If, moreover, the utilities $u_i$ are generic, the joint KKT variety over the product of simplices is finite. Since Equation (7) enforces these KKT equalities and further filters candidates by the vertex deviation inequalities, its feasible set is a finite real variety. On finite varieties, the Moment-SOS hierarchy attains exactness at some finite order, hence Equation (7) has finite convergence (returning an NE when feasible, and otherwise certifying nonexistence). $\qquad\square$

### F.4 Proofs from Section 6

**Single-player IREFGs.** First, we note that single-player IREFGs can be viewed as continuous identical-interest games (see e.g. Von Stengel & Koller (1997)), so the existence of ex-ante optima does not require concavity: $\mathcal{S}$ is compact and $u$ is continuous, hence $\arg\max_{\mathcal{S}} u \neq \emptyset$. We establish that in this setting, there is an equivalence between the definitions of concave and monotone games.

**Proposition F.2.** *A single-player IREFG $\mathscr{G}$ is monotone if and only if the expected utility $u$ is concave on $\mathcal{S}$. Moreover, $\mathscr{G}$ is strictly monotone if and only if $u$ is strictly concave on $\mathcal{S}$.*

*Proof.* (Concavity $\Rightarrow$ Monotonicity). Assume $u$ is concave. The first-order concavity inequality gives, for all $\mu, \nu$,

$$u(\mu) \ \leq \ u(\nu) + \nabla u(\nu)^\top (\mu - \nu), \qquad u(\nu) \ \leq \ u(\mu) + \nabla u(\mu)^\top (\nu - \mu).$$

Adding the two inequalities yields

$$\left(\nabla u(\mu) - \nabla u(\nu)\right)^\top (\mu - \nu) \ \leq \ 0,$$

which is exactly the definition of monotonicity. with $v = \nabla u$.

(Monotonicity $\Rightarrow$ Concavity). Assume $\langle v(\mu) - v(\nu), \mu - \nu \rangle \leq 0, \ \forall \mu, \nu \in \mathcal{S}$ (i.e. the pseudogradient is monotone). Fix $\mu, \nu \in \mathcal{S}$ and set $\gamma(t) = \nu + t(\mu - \nu)$ for $t \in [0, 1]$. Define $g(t) := u(\gamma(t))$. Then $g'(t) = \nabla u(\gamma(t))^\top (\mu - \nu)$. For $0 \leq s < t \leq 1$,

$$g'(t) - g'(s) = \left(\nabla u(\gamma(t)) - \nabla u(\gamma(s))\right)^\top (\mu - \nu) = \langle v(\gamma(t)) - v(\gamma(s)), \gamma(t) - \gamma(s) \rangle \leq 0,$$

so $g'$ is nonincreasing on $[0, 1]$. Therefore

$$u(\mu) - u(\nu) = \int_0^1 g'(t)\, dt \ \leq \ \int_0^1 g'(0)\, dt = \nabla u(\nu)^\top (\mu - \nu),$$

which is the first-order characterization of concavity, hence $u$ is concave on $\mathcal{S}$.

(Strict case). If equality in $\langle v(\mu) - v(\nu), \mu - \nu \rangle \leq 0, \ \forall \mu, \nu \in \mathcal{S}$ holds only for $\mu = \nu$, then for any $\mu \neq \nu$ and any $t \in (0, 1]$ we have

$$g'(t) - g'(0) = \langle v(\gamma(t)) - v(\gamma(0)), \gamma(t) - \gamma(0) \rangle < 0,$$

so $g'$ is strictly decreasing and $u(\mu) - u(\nu) = \int_0^1 g'(t)\, dt < \nabla u(\nu)^\top (\mu - \nu)$. This is the strict first-order concavity inequality, hence $u$ is strictly concave. The converse (strict concavity $\Rightarrow$ strict monotonicity for $\mu \neq \nu$) follows by repeating the first part with inequalities strict. $\qquad\square$

**Theorem 6.3.** *Consider a single-player IREFG $\mathscr{G}$ with utility function $u$. Let $d_0 := \max_{j,a}\{\lceil \deg(u)/2 \rceil, \lceil \deg(g_a^j)/2 \rceil, \lceil \deg(h^j)/2 \rceil\}$. Then, the following holds:*

*(i) If $\mathscr{G}$ is strictly concave/monotone, then the Moment-SOS hierarchy has finite convergence: there exists $d \geq d_0$ such that $u_d^{\mathrm{sos}} = u_d^{\mathrm{mom}} = u^*$.*

*(ii) If $\mathscr{G}$ is SOS-concave/SOS-monotone, the degree-$d_0$ Moment-SOS relaxations are exact: $u_{d_0}^{\mathrm{mom}} = u_{d_0}^{\mathrm{sos}} = u^*$, i.e., the Moment-SOS hierarchy converges at the first level.*

*Proof.* (i) Let $\mu^* \in \mathcal{S}$ be a global maximizer, so $u^* = u(\mu^*)$. Since $\mathcal{S}$ is a nonempty polyhedron and by Proposition F.2 $u$ is (strictly) concave, there exist KKT multipliers $\{\lambda_a^j\}_{j,a}$ with $\lambda_a^j \geq 0$ and $\{\nu^j\}_j$ such that

$$\nabla u(\mu^*) + \sum_{j,a} \lambda_a^j \, \nabla g_a^j(\mu^*) + \sum_i \nu^j \, \nabla h^j(\mu^*) = 0, \quad \lambda_a^j \, g_a^j(\mu^*) = 0.$$

Let $I_m$ be the $m \times m$ identity. Since $-\nabla^2 u \succ 0$ on $\mathcal{S}$, the (strictly positive) smallest eigenvalue of $-\nabla^2 u(\mu)$ is continuous in $\mu$, and the compactness of $\mathcal{S}$ implies that there exists $\delta > 0$ such that $-\nabla^2 u(\mu) \succeq \delta I_m$ for all $\mu \in \mathcal{S}$. Define the (convex) Lagrangian-type polynomial

$$G(\mu) := u(\mu^*) - u(\mu) - \sum_{j,a} \lambda_a^j \, g_a^j(\mu) - \sum_j \nu^j \, h^j(\mu).$$

Then $G(\mu^*) = 0, \nabla G(\mu^*) = 0$. Define

$$F(\mu - \mu^*) := \int_0^1 \left( \int_0^t \nabla^2 G(\mu^* + s(\mu - \mu^*)) \, ds \right) dt,$$

so that the identity holds (Helton & Nie, 2010):

$$G(\mu) = G(\mu^*) + \nabla G(\mu^*)(\mu - \mu^*) + (\mu - \mu^*)^\top F(\mu, \mu^*)(\mu - \mu^*)$$
$$= \langle \mu - \mu^*, \, F(\mu - \mu^*)(\mu - \mu^*) \rangle.$$

Since $\nabla^2 G(\mu) = -\nabla^2 u(\mu) \succeq \delta I_m$ on $\mathcal{S}$, for any $\xi \in \mathbb{R}^n$ we have

$$\xi^T F(\mu, \mu^*)\xi \geq \delta \int_0^1 \int_0^t \xi^T \xi \, ds dt = \frac{\delta}{2} \xi^T \xi.$$

Hence $F(\mu, \mu^*) \succeq \frac{\delta}{2} I_n$ for all $X \in \mathcal{S}$. Since $F(\mu, \mu^*)$ is a symmetric polynomial matrix that is positive definite on $\mathcal{S}$, the matrix polynomial version of Putinar's Positivstellensatz yields SOS-matrix polynomials $F_0, \{F_a^j\}$ and polynomial matrices $\{H^j\}$ such that

$$F(\mu, \mu^*) = F_0(\mu) + \sum_{j,a} F_a^j(\mu) \, g_a^j(\mu) + \sum_j H^j(\mu) \, h^j(\mu).$$

Multiply it on both sides by $(\mu - \mu^*)$ to obtain

$$G(\mu) = \sigma_0(\mu) + \sum_{j,a} \sigma_a^j(\mu) \, g_a^j(\mu) + \sum_j p^j(\mu) \, h^j(\mu),$$

where $\sigma_0(\mu) := \langle \mu - \mu^*, F_0(\mu)(\mu - \mu^*) \rangle \in \Sigma[\mu]$, $\sigma_a^j(\mu) := \langle \mu - \mu^*, F_a^j(\mu)(\mu - \mu^*) \rangle \in \Sigma[\mu]$, $p^j(\mu) := \langle \mu - \mu^*, H^j(\mu)(\mu - \mu^*) \rangle \in \mathbb{R}[\mu]$. Recalling the definition of $G$ and rearranging,

$$u(\mu^*) - u(\mu) = \sigma_0(\mu) + \sum_{j,a} \underbrace{\left( \sigma_a^j(\mu) + \lambda_a^j \right)}_{\in \Sigma[\mu]} g_a^j(\mu) + \sum_j \underbrace{\left( p^j(\mu) + \nu^j \right)}_{\in \mathbb{R}[\mu]} h^j(\mu).$$

Let $d = \max\{\lceil \deg(\sigma_0)/2 \rceil, \lceil \deg(\sigma_{ij})/2 \rceil, \lceil \deg(p_i)/2 \rceil\} + 1$. Then with $u^* = u(\mu^*)$, the tuple $(u^*, \sigma_0, \{\sigma_a^j + \lambda_a^j\}, \{p^j + \nu^j\})$ is feasible for the degree-$d_0$ SOS program in Equation (15), so $u_d^{\mathrm{sos}} \leq u^*$. By weak duality, we have $u_d^{\mathrm{sos}} \geq u_d^{\mathrm{mom}} \geq u^*$. Therefore, $u_d^{\mathrm{sos}} = u_d^{\mathrm{mom}} = u^*$. Conversely, choosing $y$ as the Dirac moments of $\delta_{\mu^*}$ in the moment SDP of Equation (24) gives a feasible point with value $L_y(u) = u^*$.

(ii) Recall $d_0 = \max\{d_u, d_{\mathcal{S}}\} = d_u$. Let $\mu^* \in \arg\max_{\mu \in \mathcal{S}} u(\mu)$ and set $u^* := u(\mu^*)$. Because $\mathscr{G}$ is (SOS-)concave/monotone, Proposition F.2 implies that $u$ is concave on $\mathcal{S}$. Since $\mathcal{S}$ is a nonempty

polyhedron, the KKT conditions are necessary and sufficient for optimality. Hence, for any optimal solution $\mu^*$ there exist Lagrange multipliers $\{\lambda_a^j\}_{j,a}$ with $\lambda_a^j \geq 0$ and $\{\nu^j\}_j$ such that:

$$\nabla(u)(\mu^*) + \sum_{j,a} \lambda_a^j \nabla g_a^j(\mu^*) + \sum_j \nu^j \nabla h^j(\mu^*) = 0,$$

$$\lambda_a^j g_a^j(\mu^*) = 0, \quad g_a^j(\mu^*) \geq 0, \quad \lambda_a^j \geq 0, \quad h^j(\mu^*) = 0.$$

Since $-\nabla^2 u$ is SOS-matrix and $\deg(-u) \leq 2d_0$, by (Lasserre, 2024, Theorem 3.9):

$$(-u)(\mu) - (-u)(\mu^*) - \nabla(-u)(\mu^*)^T(\mu - \mu^*) = \sigma_0(\mu),$$

with $\sigma_0 \in \Sigma[\mu]_{d_0}$. Using stationarity and linearity of $g_a^j, h^j$ (their gradients are constant), we obtain from the KKT condition:

$$\nabla(u)(\mu^*)^\top(\mu - \mu^*) = -\sum_{j,a} \lambda_a^j \nabla(g_a^j)^\top(\mu - \mu^*) - \sum_j \nu^j \nabla(h^j)^\top(\mu - \mu^*)$$

$$= -\sum_{j,a} \lambda_a^j(g_a^j(\mu) - g_a^j(\mu^*)) - \sum_j \nu^j h^j(\mu),$$

where we used $h^j(\mu^*) = 0$. Plugging this back, we have

$$u^* - u(\mu) = \sigma_0(\mu) + \sum_{j,a} \lambda_a^j(g_a^j(\mu) - g_a^j(\mu^*)) + \sum_j \nu^j h^j(\mu)$$

$$= \underbrace{\sigma_0(\mu)}_{\in \Sigma[\mu]} + \sum_{j,a} \underbrace{\lambda_a^j}_{\in \Sigma[\mu]} g_a^j(\mu) + \sum_j \nu^j h^j(\mu).$$

since $\sum_{j,a} \lambda_a^j g_a^j(\mu^*) = 0$. Thus we have the original-domain SOS certificate

$$u^* - u(\mu) = \sigma_0(\mu) + \sum_{j,a} \sigma_a^j(\mu) g_a^j(\mu) + \sum_j p^j(\mu) h^j(\mu),$$

with $\sigma_a^j(\mu) \equiv \lambda_a^j$ (nonnegative constants are SOS) and $p^j(\mu) \equiv \nu^j$. Degree bounds: $\deg(\sigma_0) \leq 2d_0$, $\deg(\sigma_a^j g_a^j) \leq 1$, $\deg(p^j h^j) \leq 1$. Thus $(u^*, \sigma_0, \{\sigma_a^j\}, \{p^j\})$ is feasible for the SOS dual Equation (15) at order $d_0$, yielding $u_{d_0}^{\text{sos}} \leq u^*$. By weak duality, we have $u_{d_0}^{\text{sos}} \geq u_{d_0}^{\text{mom}} \geq u^*$. Therefore $u_{d_0}^{\text{mom}} = u_{d_0}^{\text{sos}} = u^*$. Conversely, choosing $y$ as the Dirac moments of $\delta_{\mu^*}$ in the moment SDP Equation (24) gives a feasible point with value $L_y(u) = u^*$. $\square$

## G  EMPIRICAL EXAMPLES

In this section, we show some illustrative examples for how our proposed methods can be used to compute ex-ante optima in single-player IREFGs. We remark that we use only standard scientific computing packages in Julia, alongside an off-the-shelf SumOfSquares package (Legat et al., 2017; Weisser et al., 2019). The code is run on a PC with an AMD Ryzen 5 5600 processor and 16 GB of RAM running a 64-bit version of Windows 11, and is provided in a supplementary file.

**Example B.4.**   As a running example, we revisit Example B.4. Since the game is not multilinear (i.e. the player is absentminded), we use the standard Moment-SOS hierarchies. In particular, we run the SOS hierarchy in the Moment-SOS loop, testing the rank condition at each level until an atomic measure (i.e. a feasible maximizer) can be extracted. The program converges at truncation degree $d = 4$, returning optimal solution $(x_{11}^*, x_{12}^*) = (1, 0)$ and $(x_{21}^*, x_{22}^*) = (1, 0)$. This gives objective value $p(x^*) = 9$. The compute time required to solve this example was 0.02 seconds.

**Randomly Generated NAM-IREFG.**   We also create a procedure to randomly generate single-player IREFGs. Specifically, we seek to validate Statement (iii) of Theorem 4.1, that convergence occurs at a structure dependent level of the Moment-SOS hierarchy. For example, consider a (randomly generated) non-absentminded game $\mathscr{G}_1$ with 3 infosets and two actions per infoset, resulting in variables $x_1, x_2$ for $I_1$, $y_1, y_2$ for $I_2$, and $z_1, z_2$ for $I_3$. The payoff function for $\mathscr{G}_1$ is given by:

$$u_{\mathscr{G}_1}(x, y, z) = -4z_1 + x_2y_2 + x_2y_2z_1 - 3x_2y_1z_2 - 3x_2y_1z_1. \tag{39}$$

Due to Theorem 4.1, we expect convergence and extraction to be possible at level $d = 4$ of the hierarchy, since there are 3 infosets. Moreover, since the game is a NAM-IREFG, we can further restrict the feasible region to vertex set $\mathcal{S}_{\mathrm{vr}}$. Setting $d = 4$ in the hierarchy, we find ex-ante optimal solution $(x_1^*, x_2^*) = (0, 1)$, $(y_1^*, y_2^*) = (0, 1)$, and $(z_1^*, z_2^*) = (0, 1)$ with optimal value 1. The compute time required to solve this game was 0.06 seconds.

**Randomly Generated Absentminded IREFG.** As another example, we show that in a randomly generated absentminded game, the hierarchies empirically converge at 'reasonable' levels. Consider game $\mathscr{G}_2$ with two infosets $I_1$ and $I_2$, where the player chooses between 3 actions in each infoset. This gives variables $x_1, x_2, x_3$ in $I_1$ and $y_1, y_2, y_3$ in $I_2$. The payoff function for $\mathscr{G}_2$ is given by

$$u_{\mathscr{G}_2}(x, y) = 4x_1x_3 + 2x_2x_3y_3 - 5x_1x_2y_3 + x_1x_2y_1 - 4x_2x_3y_2y_3. \tag{40}$$

Running the hierarchies, we obtain convergence at level $d = 6$, with ex-ante optimal solution $(x_1^*, x_2^*, x_3^*) = (0.5, 0, 0.5)$ and $(y_1^*, y_2^*, y_3^*) = (0.134, 0.594, 0.272)$, giving optimal value 1. Notice that unlike the NAM case, the optimal solution is not a vertex. The total compute time required to solve this example was 0.41 seconds.

**SOS-Monotone Example.** We show experimental corroboration for Statement (ii) of Theorem 6.3. Using a technique established in Ahmadi et al. (2013), we construct a game $\mathscr{G}_{\mathrm{SOS}}$ with degree-4 polynomial utility which is SOS-convex. The polynomial is given below:

$$\begin{aligned}
u_{\mathscr{G}_{\mathrm{SOS}}}(x, y) = {} & 9.37y_2^4 + 9.37y_1^2y_2^2 + 9.37y_1^4 + 1.17x_2^2y_2^2 - 0.09x_2^2y_1y_2 + 0.94x_2^2y_1^2 \\
& + 9.37x_2^4 - 0.78x_1x_2y_2^2 - 0.52x_1x_2y_1y_2 + 0.55x_1x_2y_1^2 + 0.13x_1^2y_2^2 \\
& + 0.16x_1^2y_1y_2 + 0.13x_1^2y_1^2 + 9.37x_1^2x_2^2 + 9.37x_1^4
\end{aligned} \tag{41}$$

Even though this polynomial is quartic, we need only run the SOS hierarchy at level $d = 4$ to obtain the optimal value and extract a solution. We obtain the solution $(x_1^*, x_2^*) = (0, 1)$ and $(y_1^*, y_2^*) = (0, 1)$, with value 19.9. The compute time was $< 0.001$ seconds.

**Comparison With A Local Method.** To further illustrate the gap between local methods and SOS, we consider a small but nontrivial randomly generated absentminded game $\mathscr{G}_3$ with two infosets $I_1$ and $I_2$, where $I_1$ has 2 actions and $I_3$ has 3 actions. This gives variables $x_1, x_2$ in $I_1$ and $y_1, y_2, y_3$ in $I_2$. The payoff function for $\mathscr{G}_3$ is given by

$$u_{\mathscr{G}_3}(x, y) = 6x_1^2y_3 + 8x_1x_2y_2 - 3y_3^2 + x_1^3 + 4x_2y_1y_2 - 4x_2^2y_1^2. \tag{42}$$

Using our Moment-SOS implementation, the degree-4 relaxation is already flat and certifies the global optimum at $(x_1^*, x_2^*) = (1, 0)$, $(y_1^*, y_2^*, y_3^*) = (0, 0, 1)$, with optimal value 4. The compute time was 0.03 seconds.

As a baseline, we apply projected gradient descent (PGD) directly to the same objective over the feasible region. We define the concatenated variable $z = (x, y) \in \mathbb{R}^5$, perform gradient steps $z_{t+1} = z_t + \eta\nabla f(z_t)$ with step size $\eta = 0.02$, and after each step project the $x$- and $y$-coordinates onto their respective simplices using the standard Euclidean simplex projection. We stop when $\|z_{t+1} - z_t\|_2 < 10^{-8}$ or after 5000 iterations. Running PGD from 100 random interior initializations, $66/100$ runs converge to the global optimum, while the remaining runs converge to a distinct stationary point with payoff $\approx 2.15$. Thus, first-order methods can get trapped at suboptimal KKT points with nontrivial probability, whereas SOS returns the global solution together with a certificate at a modest relaxation degree.

