# OpenReview forum: "Solving Imperfect-Recall Games via Sum-of-Squares Optimization"
_ICLR.cc/2026/Conference — Submitted to ICLR 2026_

### Official Review · Reviewer_kjtu · 2025-10-16

**Soundness:** 3
**Presentation:** 2
**Contribution:** 2
**Rating:** 4
**Confidence:** 3

**Summary:**

This paper presents a theoretical framework for solving Imperfect-Recall Extensive-Form Games (IREFGs) by leveraging the Sum-of-Squares (SOS) optimization hierarchy. The authors establish a formal connection between IREFGs and polynomial optimization, demonstrating that the SOS hierarchy provides asymptotic convergence to ex-ante optimal strategies in single-player games and Nash equilibria in multi-player games. The key theoretical contributions include proving finite convergence under genericity assumptions and, more importantly, identifying non-absentminded and SOS-monotone games where convergence is particularly efficient; in some cases, requiring only a single semidefinite program (SDP). This work offers a principled, global optimization approach to a problem known for its computational hardness.

**Strengths:**

1. The paper provides a comprehensive and unifying framework for IREFGs, transforming a game-theoretic problem into a structured optimization one with proven convergence guarantees.

2. Moving beyond general hardness results, the paper makes substantial progress by defining and analyzing tractable subclasses like non-absentminded and SOS-monotone games. The result that SOS-monotone games can be solved with a single SDP is a notable and practical insight.

**Weaknesses:**

1. A major limitation is the pronounced gap between theoretical guarantees and practical utility. The SOS hierarchy is known to produce very large SDPs whose size grows combinatorially with the number of variables and relaxation order. The authors do not adequately address this scalability issue, nor do they demonstrate applicability beyond small toy problems, which significantly limits the method's relevance to the community.

2. The paper is almost entirely theoretical. The minimal examples in the appendix serve as proofs-of-concept but do not constitute a meaningful empirical evaluation. There is no comparison against existing baselines (e.g., LP, GD, RM) to illustrate the practical trade-offs.

3. Even for moderately sized problems, SDP solvers can face numerical instability, preventing convergence or violating the "flatness" condition required to extract a solution. The paper does not address these practical algorithmic hurdles, presenting an idealized view of the optimization process.

4. There has been works solving imperfect-recall games that LP cannot solve [1-6]. They presented the first algorithm for approximating maxmin strategies in two-player zero-sum imperfect recall games without absentmindedness and several variants.

5. SOS has been applied in imperfect-recall games recently [7]. The novelty is limited.

[1] Branislav Bosanský et al., Computing Maxmin Strategies in Extensive-form Zero-sum Games with Imperfect Recall.

[2] Jirí Cermák, Solving Imperfect Recall Games.

[3] Jirí Cermák et al., An Algorithm for Constructing and Solving Imperfect Recall Abstractions of Large Extensive-Form Games.

[4] Jirí Cermák et al., Combining Incremental Strategy Generation and Branch and Bound Search for Computing Maxmin Strategies in Imperfect Recall Games.

[5] Jirí Cermák et al., Approximating maxmin strategies in imperfect recall games using A-loss recall property.

[6] Jirí Cermák et al., Automated construction of bounded-loss imperfect-recall abstractions in extensive-form games.

[7] Vincent Leon et al., Certifying Concavity and Monotonicity in Games via Sum-of-Squares Hierarchies.

**Questions:**

1. The theory guarantees that SOS converges to a global optimum, whereas methods like LP, GD, or RM may only reach local optima (KKT points). Could you provide a concrete and non-trivial IREFG instance where your method is guaranteed to find the only correct solution, thereby clarifying its unique practical value beyond theoretical appeal?

2. The feasibility of your approach rests heavily on solving the underlying SDPs. Could you include a quantitative scaling analysis—for instance, a table relating the total number of strategic variables n and relaxation order d to the resulting SDP size (i.e., moment matrix dimension)? This would help the community gauge the realistic scope of application given current solvers.

3. In your experiments, how frequently did the SDP solver fail or fail to meet the flatness condition? What is your fallback strategy when the SOS relaxation at a tractable order d does not yield an extractable solution?

4. How can your method reliably assert the non-existence of Nash equilibria in IREFGs within bounded computation time?

5. What's the advantage of SOS against Gurobi optimizer, which guarantees global optimality upon finishing solving.

---

> ### Author Response · Authors · 2025-11-20
>
> We thank the reviewer for the thoughtful comments and questions, and for carefully reading our paper.  Below we provide our responses, and if the referee finds merit in these arguments, we would kindly ask them to reconsider their score. Due to the length of our response and the reference list, we have split it into two parts; this is Part 1.
>
> #### Scalability, SDP size, numerical issues, and non-existence certificates (W1, W3, Q2–Q4).
> We fully agree that scalability is the main limitation of any SOS-based approach. Our goal is not to claim that vanilla SDP solvers scale to very large games, but to understand what the Moment-SOS hierarchy guarantees for IREFGs and how the required relaxation degree depends on the information structure.
>
> In our formulation, with $m$ the number of behavioral variables and relaxation order $d$, the degree-$d$ relaxation has $\binom{m+2d}{2d}$ moments and a moment matrix of size $\binom{m+d}{d}$, with similar sizes for localizing matrices. Thus for realistic $m$, even moderate $d$ yields large SDPs. Our results partly mitigate this: in single-player non-absentminded games we prove exactness at degree $\ell+1$ (with $\ell$ the number of information sets), and in SOS-concave/monotone single-player games at the first degree $d_0$. Moreover, in practice finite convergence often occurs at low degree [1]. We will add a concise paragraph after Eqs. (4)-(5) giving the combinatorial counts for the degree-$d$ relaxation in terms of the number of variables and constraints, to help the community gauge the realistic scope with current solvers.
>
> Beyond our theoretical focus, there is active work on making SOS/SDP more scalable. DSOS/SDSOS relaxations trade off accuracy for much better scalability [2], and recent low-rank SDP methods reduce memory and runtime by exploiting approximate low-rank structure [3–6]. Our polynomial IREFG formulation fits into this framework, so these techniques could be combined with our degree bounds to handle larger imperfect-recall benchmarks; we will briefly mention this as a natural direction for future work.
>
> On numerical issues, our small-scale experiments did not exhibit solver failures and flatness appeared at low orders. Beyond the structured subclasses mentioned above, we only have generic finite convergence, so some levels may not yet be extractable. If computational budget allows, the natural fallback is to increase the relaxation order (or take another SVC step in the multi-player setting). When a certificate is not essential, one can also switch to first-order methods to obtain KKT points. A promising direction for future work is to obtain bounds on the quality of such KKT-based solutions in specific game classes.
>
> Regarding non-existence, infeasibility of the SOS feasibility problem at some level gives a sound certificate that no behavioral NE exists, but again we do not claim a worst-case time bound.
>
> *[1] Lasserre, Jean B. "The Moment-SOS hierarchy: Applications and related topics." Acta Numerica 33 (2024): 841-908.*
>
> *[2] Ahmadi, Amir Ali, and Anirudha Majumdar. "DSOS and SDSOS optimization: more tractable alternatives to sum of squares and semidefinite optimization." SIAM Journal on Applied Algebra and Geometry 3.2 (2019): 193-230.*
>
> *[3] Monteiro, Renato DC, Arnesh Sujanani, and Diego Cifuentes. "A low-rank augmented Lagrangian method for large-scale semidefinite programming based on a hybrid convex-nonconvex approach." arXiv preprint arXiv:2401.12490 (2024).*
>
> *[4] Han, Qiushi, et al. "A low-rank admm splitting approach for semidefinite programming." INFORMS Journal on Computing (2025).*
>
> *[5] Han, Qiushi, et al. "Accelerating low-rank factorization-based semidefinite programming algorithms on GPU." arXiv preprint arXiv:2407.15049 (2024).*
>
> *[6] Aguirre, Jacob M., et al. "cuHALLaR: A GPU Accelerated Low-Rank Augmented Lagrangian Method for Large-Scale Semidefinite Programming." arXiv preprint arXiv:2505.13719 (2025).*

---

> > ### Author Response · Authors · 2025-11-20
> >
> > *This comment continues our response from part 1.*
> >
> > #### Experiments and concrete instances where SOS helps (W2, Q1).
> > Our empirical section serves primarily as a proof-of-concept that the theoretical phenomena we prove (monotone bounds, flatness at the predicted level, etc.) do occur on concrete IREFG instances. Indeed, there are few existing benchmarks for solving generic IREFGs: LP-based methods mostly apply to 2p0s games, and while GD and RM are popular first-order local methods for equilibrium computation, their application to IREFGs was only recently explored in concurrent work to ours [1,2].
> >
> > To address the reviewer’s request, we have carried out an experiment on a small but nontrivial randomly generated single-player game explicitly illustrating the gap between local methods and SOS. Specifically, the reduced ex-ante objective is
> > $$f(x,y)= 6x_1^2y_3-4x_2^2y_1^2+8x_1x_2y_2-3y_3^2+x_1^3+4x_2y_1y_2,$$
> > with $x\in\Delta^1, y\in\Delta^2$.
> >
> > Using our Moment-SOS implementation on this problem, the degree-4 relaxation is already flat and certifies the global optimum at $x^* = (1,0)$, $y^* = (0,0,1)$, with payoff $4$; flatness and moment extraction provide an algebraic certificate that this solution is globally optimal.
> >
> > As suggested by the referee, we also apply projected gradient descent (PGD) directly to the same objective over the feasible region. We define the concatenated variable $z = (x,y) \in \mathbb{R}^5$, and perform gradient steps $z_{t+1} = z_t + \eta \nabla f(z_t)$ with step size $\eta = 0.02$, and after each step project the $x$- and $y$-coordinates onto their respective simplices using the standard Euclidean simplex projection. We stop when $\|z_{t+1} - z_t\|_2 < 10^{-8}$ or after $5000$ iterations. Running PGD from $100$ random interior initializations, $66/100$ runs converge to this optimum, while the remaining runs converge to a distinct stationary point with payoff $\approx 2.15$. Thus, local methods can get trapped at suboptimal KKT points with nontrivial probability, whereas SOS returns the global solution together with a certificate.
> >
> > In the revised version, we will include this example in a concise form in the experiments section, explicitly comparing the merits of both approaches.
> >
> > *[1] Anagnostides, Ioannis, et al. "Convergence of regret matching in potential games and constrained optimization." arXiv preprint arXiv:2510.17067 (2025).*
> >
> > *[2] Tewolde, Emanuel, et al. "Decision Making under Imperfect Recall: Algorithms and Benchmarks." Uncertainty in Artificial Intelligence (UAI). 2025.*
> >
> > #### Relation to prior imperfect-recall and SOS work (W4, W5).
> > We thank the reviewer for providing additional references [1-7] and will include them as related work in subsequent versions of the paper. We remark that references [1-6] focus on the 2p0s case, while our results extend to multiplayer general-sum IREFGs using a drastically different approach. Future work comparing and even combining these approaches would certainly be of interest.
> >
> > While SOS has indeed been applied to IREFGs, the paper cited uses SOS for a very different purpose. In particular, [7] uses SOS techniques to verify concavity and monotonicity in polynomial games, and they do not deal at all with equilibrium computation.
> >
> > #### Advantage of SOS vs Gurobi (Q4).
> > The problems we study are nonconvex polynomial programs over continuous strategy spaces, which fall outside the classes where solvers like Gurobi provide global guarantees. The Moment-SOS hierarchy, by contrast, constructs a sequence of convex SDPs whose values form monotone bounds, converge (asymptotically and often finitely) to the global optimum, and yield explicit certificates of optimality or infeasibility via flatness and dual infeasibility. The trade-off is that these certificates may require higher relaxation levels, reflecting the inherent hardness of the underlying problems.

---

> > > ### Comment · Reviewer_kjtu · 2025-11-25
> > >
> > > Thank you for your response. I note that the main contribution of this paper lies in its theoretical, which falls somewhat outside the scope of my research. Consequently, I have maintained my rating while reducing my confidence.

---

### Official Review · Reviewer_hZLc · 2025-10-30

**Soundness:** 3
**Presentation:** 2
**Contribution:** 2
**Rating:** 6
**Confidence:** 2

**Summary:**

The paper studies imperfect-recall games and whether Moment Sum-of-Squares (SOS) hierarchy provides computational benefits to some subclasses of those games. The paper shows that in non-abstentminded games, the Moment-SOS hierarchy converges in finite time even with single instantiation of hierarchy both in single-player and multi-player settings. In absent-minded games, the method converges asymptotically to a Nash equilibrium (or it proves it's nonexistence) with multiple instantiations of the hierarchy. The authors then define SOS-certifiable counterparts of concave and monotone games and show that in single-player SOS-monotone games, the Moment-SOS converges using only a single SDP.

**Strengths:**

* Improves one of the hardness-results for single-player imperfect recall games.
* Deepens the connection between Moment-SOS and the polynomial games, which was proposed in [1].
* Provides practical usage of Moment-SOS for imperfect recall games and the behavior of Moment-SOS for some subclasses of those game.

**Weaknesses:**

* Most of the results seem to directly follow from the definition of imperfect recall games as a polynomial game, the construction provided in [1] and the behavior of Moment-SOS.
* Without the prior knowledge the concepts introduced in Section 2.2 would be really difficult to follow, which authors probably realized and provided 2 works that delve more into those concepts.
* The clashing notation of SOS and games makes the paper difficult to follow at first. Some notational details are not defined like $\mu_{-i}$ (specifically the use of the negative subscript index) or the sum of multi-indices $\alpha + \beta$,
* Little empirical evidence supporting the theory.

**Questions:**

1. You have defined SOS-concave games, but have not used this property anywhere. Are there any computational benefits of those games (except the fact, they have a behavioral Nash)?
2. The SVC method seems to follow similar procedure as the double oracle algorithm. Is there some connection between those methods?

---

> ### Author Response · Authors · 2025-11-20
>
> We are grateful to the reviewer for the thoughtful comments and questions, and for taking the time to carefully read our paper. We will respond to each of your concerns separately below.
> #### Most Results follow from existing polynomial game construction + Moment-SOS behavior.
> While the Moment-SOS approach is well-studied for polynomial problems, we leverage structural properties that are specific to IREFGs to derive improved convergence guarantees. Concretely, we specialize this approach in three game-dependent ways:
>
> (a) Classical finite-convergence results assume generic data (both utilities and constraints). Our feasible region is a product of simplices with structured (hence non-generic) constraints. We assume genericity only of the utility and adapt the finite-convergence arguments to this game-theoretic geometry; this is not a black-box application.
>
> (b) We show that non-absentmindedness in single-player IREFGs implies convergence of the hierarchy at level $\ell+1$, where $\ell$ is the number of information sets. This yields an explicit, game-structural degree bound, going well beyond what generic polynomial optimization theory would provide.
>
> (c) We define SOS-concave/SOS-monotone IREFGs, for which the hierarchy converges in a single step. These SOS variants can be checked by a single SDP, giving a tractable surrogate for the “good curvature” regime where concavity or monotonicity are NP-hard to verify.
>
> Our analysis is tightly tied to the fact that the underlying polynomial problems come from games, and are not just a black-box application of generic SOS results. We also view our work as opening a natural direction: identifying and analyzing other subclasses of IREFGs with tractable guarantees.
>
> #### Difficulty of Section 2.2 without prior knowledge.
> We agree that Section 2.2 is dense. The step-by-step derivation of the hierarchy is already in Appendix D, but in future revisions we will briefly summarize in Section 2.2 the key notions we rely on while keeping the main text self-contained.
>
> #### Clashing notation and missing details.
> We will clarify in the main text that $\mu_{-i}$ denotes “all components except player $i$”, and that sums of multi-indices like $\alpha+\beta$ are taken componentwise. We will also do a pass to ensure that SOS notation and game-theoretic notation are introduced cleanly and used consistently.
>
> #### Little empirical evidence.
> Our work is primarily theoretical, but we agree that some empirical illustrations help ground the theory. Appendix G already contains several single-player IREFG examples where we compute successive SOS relaxations, identify the level at which flatness occurs, and verify that this matches our structural bounds. In addition, we ran a new experiment on a single-player IREFG, comparing Moment-SOS to projected gradient descent (PGD). The degree-4 SOS relaxation certifies the global optimum, whereas PGD (from 100 random interior initializations) converges to the same optimum in 66/100 of runs and to a distinct suboptimal stationary point in the remaining. This illustrates that local methods can get stuck at suboptimal KKT points even in small instances, while SOS returns a globally optimal solution.
>
> In the revision, we will explicitly flag the Appendix G examples and summarize this PGD vs. SOS comparison in the main text.
>
> #### Computational benefits of SOS-concave games (Q1).
> In the single-player case, we show in Proposition F.2 that concavity and monotonicity coincide for ex-ante utilities, so SOS-concave $\Leftrightarrow$ SOS-monotone. Thus, Theorem 6.3 can be stated directly for SOS-concave/SOS-monotone single-player IREFGs: any SOS-concave game automatically satisfies the SOS-monotone condition used in the proof and enjoys degree-$d_0$ convergence. We will edit the statement of Theorem 6.3 accordingly and make this equivalence explicit in the main text.
>
> #### Connection between SVC method and double oracle (Q2).
> Both SVC and double oracle (DO) alternate between solving a reduced problem and enforcing best responses, but they proceed in different directions. SVC works “top-down”: it solves a KKT system, checks whether the resulting point is an NE via best-response conditions, and if not, adds a cut that rules out that KKT point. DO works “bottom-up”: it starts from a small subgame, computes an NE of that subgame (typically via LP), then adds best-response actions to enlarge the subgame. Thus, SVC iteratively eliminates non-equilibrium KKT points, whereas DO iteratively builds up an NE. SVC provably converges to an NE in any generic IREFG, whereas similarly general convergence results for DO are not known beyond special cases (e.g., two-player continuous zero-sum games). That said, DO-based methods often work well in practice on large games, and it would be interesting to study them in IREFGs in future work.

---

> > ### Comment · Reviewer_hZLc · 2025-11-26
> >
> > I would like to thank the reviewer for their response. I forgot to include the reference for [1] in my original review, so I am adding it now.
> >
> > I was mildly positive about the paper and this still remains even after the response. I still believe the main idea of using Moment-SOS for imperfect recall games and showing which types of those games enable faster convergence is interesting. However, I will not improve the rating, as even after the response, I feel that most of the results follow from the Moment-SOS hierarchy.
> >
> > To address the points in authors response:
> > I agree with the authors that they bring some contribution to the field. I am not very familiar with Moment-SOS literature, but to me, it seems that the claims made by the authors are based on results known in Moment-SOS and only demonstrate that some games correspond to these known classes. I am not trying to undermine the contribution in that, but I believe it is worth pointing out. If there are some results that are new even to the Moment-SOS hierarchy, the authors should stress this more in the paper. Moreover, as pointed out by reviewer Zpxn, it is unclear whether any reasonable games satisfy the SOS concavity and SOS monotonicity. As a result, I am not sure whether these results are useful.
> > Because of these reasons I will not change my score.
> >
> > [1] Jiawang Nie and Xindong Tang. Nash equilibrium problems of polynomials. Mathematics of Operations Research, 49(2):1065–1090, 2024.

---

### Official Review · Reviewer_suQa · 2025-10-31

**Soundness:** 3
**Presentation:** 3
**Contribution:** 2
**Rating:** 2
**Confidence:** 3

**Summary:**

The paper investigates the use of the sum of squares (SoS) relaxation hierarchy as a tool to solve imperfect-recall games. The paper uses various results from the SoS literature that, when carried over to the game setting, yield results regarding the convergence of the SoS hierarchy to equilibria in games.

**Strengths:**

The paper is well written and clear. I find the idea of using SoS as a relaxation for imperfect recall games very interesting; indeed, it is an idea that I myself have toyed around with a bit, albeit without much progress.

**Weaknesses:**

The paper should also cite and compare to [1, 2], which covers timeable two-player zero-sum imperfect-recall games (team games). The techniques used in these papers, although not SoS, are basically "lift-and-project"-style algorithms, and have the same flavor of complexity that depends on a "degree-like" parameter, which the papers characterize.

My main criticism is that the paper feels a bit preliminary. The results follow mostly from basically restating known results in SoS land once one has expressed the program of finding an equilibrium as a polynomial feasibility program. It seems that one could have written this paper about just about any class of problems that can be reduced to polynomial optimization, and that's a lot of problems. What makes *games* special here? I was hoping to see more analysis of special things that happen in imperfect-recall games.

See *Questions* below for more specific ideas/questions about this.

My rating is negative, but again, I think that the idea is interesting. I think that with further development this can be a very strong paper.


[1] BH Zhang, G Farina, T Sandholm (ICML 2023) "Team Belief DAG: Generalizing the Sequence Form to Team Games for Fast Computation of Correlated Team Max-Min Equilibria via Regret Minimization"

[2] BH Zhang, T Sandholm (AAAI 2022) "Team correlated equilibria in zero-sum extensive-form games via tree decompositions"

**Questions:**

1.  Do you have some understanding of *at what degree* the relaxation becomes tight in general? For example, [1, 2] relate the size (dimension) of their lifted strategy set to the information structure of the game, which is kind of like the degree of the SoS relaxation. Can you make a similar statement beyond their setting of timeable games? What would that look like?

2. The paper introduces SoS-concave and SoS-monotone games, but doesn't seem to do much with them. I am left with more questions that I started with. For example, what sorts of games are SoS-concave or SoS-monotone? Why should I care about this class of games, beyond "they are the class of games for which Theorem 6.3 can be proven"?

---

> ### Author Response · Authors · 2025-11-20
>
> We thank the reviewer for their careful reading and detailed comments. We are encouraged that they find the problem interesting and that this is “an idea they have toyed around with a bit, albeit without much progress”, which we take as evidence that the problem is both nontrivial and relevant to experts in the area. At the same time, the reviewer has assigned a score of 2, on the grounds that the paper “feels a bit preliminary” and that: “the results follow mostly from basically restating known results in SOS”. The reviewer asks: “What makes games special here? I was hoping to see more analysis of special things that happen in imperfect-recall games.” We respectfully believe this assessment understates the contributions of our work. Below we clarify our key points, and if the referee finds merit in these arguments, we would kindly ask them to reconsider their score.
>
>  #### On the novelty of bringing SOS hierarchies to equilibrium computation in EFGs.
> This work is the first systematic use of Moment–SOS hierarchies to compute equilibria in EFGs, especially IREFGs. While it is well known in the game theory community that IREFGs can be recast as polynomial games, and there is work in optimization applying SOS hierarchies to polynomial games, the latter mainly considers contrived examples. Our first contribution is to make this connection explicit and utilize it in a structured way for a broad and natural class of games. It opens the door to treating equilibrium computation in IREFGs with the full SOS toolbox.
>
> #### On IREFG-specific analysis of the SOS approach.
> Rather than just restating generic SOS results, we exploit structural properties specific to IREFGs:
>
> *(a) Finite convergence with structured simplex constraints.*
>
> Classical finite-convergence results for SOS hierarchies assume generic data  (both utilities and constraints). Our feasible region is a product of simplices with structured constraints, so the constraint polynomials are non-generic. We assume genericity only of the utility and adapt the finite-convergence arguments to this particular game-theoretic geometry; this is not a black-box application.
>
> *(b) Non-absentmindedness implies convergence at level $\ell+1$.*
>
> We show that the structural property of non-absentmindedness in single-player IREFGs implies convergence of the hierarchy at level $\ell+1$, where $\ell$ is the number of information sets. This yields an explicit, game-structural degree bound, going well beyond what generic polynomial optimization theory would provide.
>
> *(c) SOS-concave/SOS-monotone games: one-step convergence.*
>
> We define SOS-concave/SOS-monotone IREFGs, as games for  which the hierarchy converges in a single step. The definition is motivated by the SOS relaxation itself (games that are “easy” for the hierarchy), but it is still in game-theoretic terms. These SOS variants can be checked by a single SDP, so they form a tractable surrogate for the “good curvature” regime in which concavity or monotonicity are NP-hard to verify. As the reviewer and we both note, it remains an open and interesting question to identify natural subclasses of IREFGs that fall into this category; we are actively exploring this direction.
>
> These points address the referee’s questions “what makes games special?” and “at what degree the relaxation becomes tight”: we show how special structural properties of IREFGs translate into concrete convergence guarantees and degree bounds for the SOS hierarchy. Our analysis is directly derived from the game structure, and is not just a black-box application of generic SOS results.
>
> #### On missing related work [1,2] on team games.
> The reviewer mentions lift-and-project-type constructions [1,2] that yield linear representations parametrized by the treewidth of the game tree. These methods are interesting, but they are tailored to timeable two-player zero-sum imperfect-recall games. In contrast, our SOS-based approach applies to arbitrary multi-player IREFGs. Comparing the two methods in the special subclass considered in the cited work (e.g., in terms of level of convergence of SOS vs. treewidth) is an interesting question. In [2], the size of the lifted polytope is parameterized by the number of tree nodes, the treewidth, and a measure of uncommon external information; [1] has analogous parameters for the TB-DAG. Our NAM single-player result shows that, for a different class of games, SOS tightness can be bounded in terms of the number of information sets, which plays a role somewhat analogous to such structural parameters. Extending this type of parametrization to general multi-player IREFGs is natural. However, from the SOS side, the interplay between treewidth and SOS hierarchies is relatively unexplored: treewidth is mostly used for symmetry reduction and not as a parameter controlling convergence levels. We believe this idea warrants a dedicated study, and will highlight this more explicitly as a direction for future work.

---

> > ### Comment · Reviewer_suQa · 2025-11-23
> >
> > I suppose here is another way of saying one of my main issues: as researchers interested in equilibrium computation, our ultimate goal is to design algorithms that efficiently compute equilibria. What interests me most in this paper, therefore, is discussion of when SoS-type algorithms can lead to *more efficient* computation of equilibria than known before. This is why I asked about degree bounds: one way of showing that the algorithm is efficient is to show that it converges at low degree. And the paper does attempt to provide this, but I have several issues:
> >
> > 1. On SoS-concavity/monotonicity: It is worth noting that, while verifying monotonity might be hard in general, there are many known algorithms (e.g., extra-gradient ascent, or ellipsoid) that find Nash equilibria of monotone games (and, more generally, solutions to monotone variational inequalities). Thus, one way to circumvent the hardness of verifying monotonicity is to simply 1) run one of these algorithms, 2) check whether it outputs an equilibrium, and 3) if it does not, assert that the game is non-monotone. (Notice that this does not contradict the hardness of verifying monotonicity, because the algorithm might, by coincidence, find an equilibrium even when the game is non-monotone.) See e.g. [3] and citations therein. Thus, *even if* one were to characterize precisely what games are SoS-monotone/concave, it still would not lead to any better algorithms. (I probably should have mentioned this point in my review, but didn't think of it at the time. Apologies.)
> >
> > 1.   There are many characterizations in the paper of the form "the SoS hierarchy converges in the limit/at some large degree". However, the degree bounds, even when they have explicit bounds, are useless when it comes to designing better algorithms: the only explicit degree bound provided is $\ell+1$ for the one-player non-absentminded case, but this only leads to a program of size something like $|\mathcal H|^{O(\ell)}$---if given this much time, I could also compute the optimal strategy by simply enumerating every single pure strategy, of which there are at most $|\mathcal A|^{\ell}$, since in this case the optimal strategy is guaranteed to be pure!
> >
> > Regarding the comparison to [1, 2] specifically: Another difference is that they allow mixed strategies, whereas the present paper is about behavioral equilibria. So, except in the single-player timeable case, the settings are rather different. But it is worth noting that in the single-player timeable case, their results seem to dominate those in the present paper, since their upper bounds are always (weakly) better than the trivial  $|\mathcal H|^{O(\ell)}$ that you would get from your result.
> >
> > A different (orthogonal) way of demonstrating the utility of your method would be to run experiments showing that it either finds equilibria where no other methods can, or finds equilibria more efficiently. I appreciate the experiments in the appendix, but they are too small scale/toy to draw any conclusions of this sort. A proper experiment here should also compare to off-the-shelf nonconvex optimizers like that implemented by Gurobi.
> >
> > I think we are in agreement that investigating when SoS actually leads to better algorithms is an interesting future direction, and I urge the authors to pursue it. I suppose the disagreement is that I think that going in this direction is *necessary* to strengthen the paper, for the current results are not sufficient in my opinion. Other reviewers seem to share my concerns with varying degrees of strength. I am optimistic that such results could lead to a much stronger paper, but, as my core opinion has not changed, I maintain my score.
> >
> > [3] A Nemirovski, S Onn, UG Rothblum (Math of OR 2010), "Accuracy certificates for computational problems with convex structure"

---

> ### Author Response · Authors · 2025-12-02
>
> Computationally speaking, there are to our knowledge no methods that can compute equilibria in generic IREFGs. Hence, seeking *more efficient methods* as suggested by the reviewer is a moot point. We further respond to the reviewer’s comments below:
>
> #### On SOS concavity/monotonicity.
> We disagree that one can ‘circumvent’ verifying SOS concavity/monotonicity by running an algorithm and then ‘check whether the algorithm outputs an equilibrium’. While this is certainly viable for non-absentminded (i.e., multilinear) games, checking whether a given point is an NE is intractable in the general case. Even if the algorithm were to converge and were to be verified, this does not say anything about the class that the game belongs to, giving no meaningful structural insight.
>
>
> #### On comparison with enumeration.
> In the single-player non-absentminded case, we disagree that our method is comparable to “simply enumerating every single pure strategy”. Our bound $|\mathcal H|^{O(\ell)}$ is a bound on the size of the SDP formulation, not on runtime or convergence. Enumeration performs combinatorial search over $|A|^{\ell}$ pure policies, whereas the SOS relaxation is solved by convex optimization over a continuous feasible set and certifies optimality once solved. Similar exponents in the size bounds therefore do not make enumeration an equivalent alternative.
>
> Moreover, general results for $0-1$ polynomial optimization guarantee exactness of Lasserre's hierarchy only at order $n+v$ [4], where $n$ is the number of binary variables and $v$ depends on the constraint degrees. In contrast, by exploiting the IREFG structure we obtain exactness already at order $\ell+1$, which can be substantially smaller than $n+v$.
>
> #### On sizes of LP vs SDP.
> Regarding [1,2], we argue that their results are not directly comparable to ours, both in setting and in solution concept:
> * They study two-player zero-sum team games with timeable information structure, a subclass of non-absentminded games, and compute team-correlated (mixed) strategies. Using a junction-tree construction, they obtain an LP of size $O(N W^{w+1})$ in [2].
> * We study multi-player, arbitrary IREFGs (including absentminded games) and focus on behavioral equilibria. Our approach is based on Lasserre's SOS hierarchy, for which we derive bounds on the relaxation degree.
>
> If one insists on a direct comparison, the only real overlap is the single-player non-absentminded "team vs. nature" case, where both frameworks reduce to optimizing over pure plans in $\\{0,1\\}^n$. By Wainwright's theorem for $0-1$ multilinear systems [5], in the canonical formulation each infoset constraint $\sum_{a\in A(I)} \mu_{I,a} = 1$ forces the treewidth $t$ to satisfy $t \ge \max_I |A(I)| - 1$, so any tree-decomposition-based method must pay at least this price. Lasserre's guarantee then yields exactness at order $t+v$ (for some $v$ depending on the constraint degrees), whereas our SoS result guarantees exactness at order $\ell+1$, independent of $\max_I |A(I)|$. In regimes with many actions per infoset but short information depth (large $\max_I |A(I)|$, small $\ell$), our $\ell+1$ bound can therefore be more favorable than bounds that scale with treewidth, and the factor $W^{w+1}$ in [2] does not dominate our $|\mathcal H|^{O(\ell)}$ scaling.
>
> #### On comparison with Gurobi.
> Regarding the suggestion to compare directly to Gurobi, our method targets a different regime. (i) Gurobi’s global guarantees apply to quadratic (and quadratically constrained) models, so the higher-degree polynomial payoffs in our IREFGs must first be manually reformulated as quadratic programs with auxiliary variables. (ii) The quality of the resulting solutions then depends heavily on this reformulation and problem structure. In contrast, our SOS formulation works natively on the polynomial payoffs from the game tree, without any hand-crafted quadratic re-encoding.
>
>
> *[4] Lasserre, Jean B. "An explicit equivalent positive semidefinite program for nonlinear 0-1 programs." SIAM Journal on Optimization 12.3 (2002): 756-769.*
>
> *[5] Wainwright, Martin J., and Michael I. Jordan. Treewidth-based conditions for exactness of the Sherali-Adams and Lasserre relaxations. Technical Report 671, University of California, Berkeley, 2004.*

---

### Official Review · Reviewer_Zpxn · 2025-11-01

**Soundness:** 4
**Presentation:** 3
**Contribution:** 4
**Rating:** 8
**Confidence:** 2

**Summary:**

This submission employs sum-of-squares (SOS) hierarchies for computing Nash equilibrium (working on behavioral strategies) in imperfect-recall extensive-form games (IREFGs), where players may forget information previously available to them during gameplay, and computation of solution concepts are harder than the perfect recall case, even with a single-player. They show that for single-player IREFGs, SOS hierarchy converges asymptotically to the maximum possible (ex-ante optimal) utility in the game, that the convergence is finite under a genericity assumption, and that in games without absentmindedness (no information set gets visited multiple times during gameplay), the convergence can be bounded by the number of infosets in the game. Similarly, the authors show that for multi-player IREFGs, multiple instantiations of the SOS hierarchy can be used to converge to behavioral Nash equilibria if its exists (again in finite steps under certain assumptions), and certify non-existence otherwise. Lastly, they define the SOS-monotone and SOS-concave IREFGs, which are subsclasses of IREFGs where the computation of their method becomes more tractable.

**Strengths:**

1) The tackles an important problem that has been receiving increasingly popular attention. The authors do an excellent job covering some of the recent (and more classic) related literature on imperfect recall games, and motivate their setting by discussing their applications in solving large games via abstractions, modeling team games, and for safety & security. The existing negative results are clearly presented and therefore it is very clear where the contributions of this paper fits in the literature.

2) The theoretical analysis is rigorous and aptly exploits the connection between imperfect recall games and polynomial optimization to make full use of the sum of squares hierarchies. The assumptions needed for their various positive results are clearly given.

3) The paper is generally well-written with a clear organization. The sections nicely connect to and build on top of each other.

**Weaknesses:**

1) At times the paper becomes incredibly dense with overwhelmingly many definitions and notation. I found sections 2.2 and 3 particularly difficult to follow for someone without significant experience in sum of squares / polynomial optimization. While this is somewhat inevitable due to the many concepts required for defining both imperfect-recall extensive-form games and SOS hierarchies and the limited space, it does end up decreasing the accessibility of the paper. I would suggest the authors to use the 10th content page they get during the rebuttal process to provide intuitions for the many definitions presented while introducing SOS hierarchies.

2) Pretty much all proofs are deferred to the appendix. Once again, there is not much the authors can do about this due to the page limit, but providing proof sketches for the main theorems can be extremely helpful in understanding and appreciating the technical novelty required for the proofs, which is currently not clear without looking at the appendix.

3) While the definitions for SOS-concave and SOS-monotone IREFGs are clear and lead to nice tractability results, the authors do not give any examples of natural games that would fall into this category. Motivating the applicability of these subclasses by discussing settings in which they can arise (especially in relationship to the settings discussed in the introduction for motivating IREFGs) would significantly strengthen section 6.

Minor:
- abstract: SDP acronym used without being introduced.
- l136: $\mathcal{H} \notin \mathcal{Z}$ -> $\mathcal{H} \setminus \mathcal{Z}$
- l152: "a distribution $\mu(...)$" -> should be $\mu_i$
- l175 "i.e., no player can profitably deviate from $\mu^*$ at any of their information sets." This sounds more like the description of EDT equilibria, since it can be interpreted as each player being able to change their strategy at only a single information set. It would be nice to make it clear that multi-information set deviations are also allowed.
- line219: why not just use $\pi$ for nodes in EFGs to begin with instead of overloading $h$?
- l279 "optmality" typo
- l280 "active gradients" repeated
- l412 "Leverage" -> "Leveraging"

**Questions:**

1) The KKT conditions of each individual player’s optimization problem exactly correspond CDT equilibria, a solution concept weaker than Nash [Tewolde 2023,2024]. Does (ii) of Theorem 4.1 imply that under the genericity assumption, all CDT equilibria (KKT points) are ex ante optimal? Also, can the verify-cut steps in Section 5 be removed to compute CDT equilibria (that are not necessarily Nash)? If so, what can we say about the improvements to the convergence guarantees?

2) l105 briefly mention perfect-recall refinement, which are an important concept for computing the value of recall of IREFGs (see Berker et al. 2025, "Value of Recall in Extensive-Form Games"). It seems a straightforward adaptation of the SOS-based methods introduced in your paper can be used for computing the value of recall in games as well through the naive approach of computing the ex ante optimal utility of the IREFG, and then solving the perfect-recall refinement exactly (which is at least easy for single-player or 2p0s games), and compare these utilities. Do you think there might be ways to apply SOS hierarchies to compute value of recall in ways more efficient than this naive approach (i.e. without computing the ex-ante optimal utilities directly)?

---

> ### Author Response · Authors · 2025-11-20
>
> We are grateful to the reviewer for their support of our paper, and for the detailed and insightful comments. We will respond to each of your comments separately below.
>
> #### Density of notation and accessibility of Sections 2.2 and 3.
> We agree that Sections 2.2 and 3 are dense, especially for readers without prior exposure to Moment-SOS or polynomial optimization. The step-by-step derivation of the hierarchy is already in Appendix D, but in future revisions we will briefly summarize in Section 2.2 the key notions we rely on (moments, localizing matrices, flatness) while keeping the main text self-contained.
>
> #### Proof sketches for main theorems.
> We agree and will revise the main text accordingly. For the central results (Theorems 4.1, 5.2, and 6.3), we will add short proof sketches in the main text.
>
> #### Examples of natural SOS-concave and SOS-monotone IREFGs.
> Our goal in Section 6 is not to argue that SOS-concave or SOS-monotone IREFGs capture a broad “natural” class of imperfect-recall games, but to isolate a structural subclass where both equilibrium analysis and the Moment-SOS hierarchy behave especially well. We will make this role clearer in the revision. Concretely, these games are SDP-checkable analogues of Rosen-concave and Rosen-monotone games: they imply concavity/monotonicity but not conversely, and membership can be certified by a single SDP. This is useful because checking concavity or monotonicity of general polynomials is NP-hard, whereas the SOS versions give a tractable surrogate notion of “nice curvature”. We do not claim that these assumptions hold in a wide “natural” family of IREFGs; they are deliberately strong, much like convexity or strong monotonicity in classical settings. The payoff is that, under these conditions, equilibrium existence is guaranteed and the Moment-SOS hierarchy has good convergence.
>
> #### Minor comments and typos.
> We thank the reviewer for the detailed minor suggestions; we will correct them all in the revision. We will also sweep the draft for any other typos while doing these edits.
>
> #### CDT equilibria, KKT, and verify-cut (Q1).
> We thank the reviewer for highlighting the connection to CDT equilibria in [1,2]. In these works, CDT equilibria are exactly profiles where each infoset plays a KKT point of its local optimization problem, so our KKT formulation is very close in spirit.
>
> Theorem 4.1(ii) does not imply that all CDT equilibria/KKT points are ex-ante optimal; it only guarantees that the KKT-based Moment-SOS hierarchy converges, generically in finite degree, to a global optimal KKT point, while other suboptimal KKT points (local maxima or saddles) may still exist.
>
> In the multi-player case, if we drop the verify-cut loop and only enforce each player’s KKT conditions, then the hierarchy becomes a finite-convergence procedure under genericity for CDT equilibria: at some finite level it either finds a CDT equilibrium or certifies infeasibility.
>
> *[1] Tewolde, Emanuel, et al. "The computational complexity of single-player imperfect-recall games." arXiv preprint arXiv:2305.17805 (2023).*
>
> *[2] Tewolde, Emanuel, et al. "Imperfect-recall games: Equilibrium concepts and their complexity." arXiv preprint arXiv:2406.15970 (2024).*
>
> #### Value of recall/perfect-recall refinement (Q2).
> In [3], for a given solution concept $\mathrm{SC}$ and an imperfect-recall game $G$ with coarsest perfect-recall refinement $R_1(G)$, the value of recall is $\mathrm{VoR}_{\mathrm{SC}}(G)=\frac{u_1(\mathrm{SC}(R_1(G)))}{u_1(\mathrm{SC}(G))}$, i.e., the ratio between Player 1’s utility with and without recall.
>
> In our setting, $u_1(\mathrm{SC}(G))$ can be approximated by the Moment-SOS hierarchy whenever $\mathrm{SC}$ is specified via an optimization problem over behavioral strategies. For the perfect-recall refinement $R_1(G)$, standard extensive-form techniques (e.g., sequence-form LP or dynamic programming) can compute $u_1(\mathrm{SC}(R_1(G)))$ exactly in the single-player and 2p0s settings considered in [3]. Taking their ratio yields a straightforward SOS-based approximation of the value of recall.
>
> Compared to the naive procedure, we do not currently see a clear SOS formulation that is asymptotically cheaper. One could, in principle, set up a single polynomial optimization problem with variables for both $G$ and $R_1(G)$ and encode the ratio directly, but the resulting SOS relaxations would have essentially the same size and degree as solving the two problems separately.
>
> Designing specialized SOS formulations tailored specifically to compute value-of-recall efficiently is an interesting direction for future work, but it lies beyond the scope of the present paper.
>
> *[3] Berker, Ratip Emin, et al. "The Value of Recall in Extensive-Form Games." Proceedings of the AAAI Conference on Artificial Intelligence. Vol. 39. No. 13. 2025.*

---

### Author Response · Authors · 2025-12-02

**Dear Area Chair,**

We thank you and the reviewers for the time and effort devoted to our submission. Given the increased workload in view of the data leak, we summarize here the most pertinent points regarding the contributions and the discussion with the referees.

**Overview of our contributions.**
This paper studies equilibrium computation in extensive-form games (EFGs), a powerful framework for sequential decision making. Within EFGs, we focus on:

1. **Imperfect-recall games**, where players may forget parts of their history or previously acquired information – a natural and important class; and
2. **Behavioral strategies**, i.e., independent randomization at each decision point, which is the standard strategy formalism in this setting (as opposed to mixed strategies).

Our main contribution is to develop a **moment/sum-of-squares global optimization framework** for computing Nash equilibria in behavioral strategies. Our results are best understood via a two-way split:

* **Single-player setting.** Vanilla Moment/SOS theory applies immediately but only yields asymptotic convergence. We exploit game structure to obtain:

  1. In the special case of *non-absentminded* games, finite convergence at a level of the hierarchy depending explicitly on the number of information sets.
  2. A *structurally defined class* of games (SOS-monotone) where convergence occurs at the *first level*, giving a polynomial-time algorithm for equilibrium computation.
  3. *Finite convergence for almost all games* via a KKT-based hierarchy.

* **Multi-player setting.** We design an algorithm that alternates between solving SOS relaxations and a cutting-plane–type procedure, and prove that it *terminates in finite time for almost all games*.

The scores we received are:

* Reviewer Zpxn: **8** (confidence 2)
* Reviewer suQa: **2** (confidence 3)
* Reviewer hZLc: **6** (confidence 2)
* Reviewer kjtu: **4**  (confidence 3)

We are grateful that two reviewers placed the paper clearly above the acceptance threshold (6 and 8).

On the other hand, we believe that Reviewer suQa’s assessment with  *score of 2* does not accurately reflect the technical content of the paper or the reviewer’s own written comments.

In their *initial review*, Reviewer suQa notes that the problem is interesting and remarks that this is “an idea they have toyed around with a bit, albeit without much progress,” which we take as evidence that the problem is both nontrivial and relevant to experts. At the same time, they describe the paper as “preliminary,” suggest that the results “mostly” restate known SOS facts, and ask “what makes games special here?”. They ask to compare our work with [1,2] that use “lift-and-project methods” in a very special setting far from ours. Finally, in their *second response*, the reviewer writes: “What interests me most in this paper, therefore, is discussion of when SOS-type algorithms can lead to more efficient computation of equilibria than known before.”

We now paste below their main comments and our responses:

#### Comment 1: “There are many characterizations in the paper of the form ‘the SoS hierarchy converges in the limit/at some large degree’. However, the degree bounds, even when they have explicit bounds, are useless when it comes to designing better algorithms: the only explicit degree bound provided is for the one-player non-absentminded case, but this only leads to a program of size something like ---if given this much time, I could also compute the optimal strategy by simply enumerating every single pure strategy, of which there are at most , since in this case the optimal strategy is guaranteed to be pure!”

Response: The reviewer writes that our degree bounds are “useless” for algorithm design and argues that one might as well enumerate all pure strategies. We respectfully disagree. It is well known that central optimization and decision problems in imperfect-recall EFGs are *computationally hard*. In the absence of additional structure or approximation, one cannot expect sub-exponential worst-case behavior in natural size parameters (numbers of information sets, pure strategies, etc.). Any exact algorithm that solves the full class—including brute-force enumeration—is necessarily exponential in the worst case. Our explicit bound for the non-absentminded single-player case is another such worst-case exponential bound, but one grounded in a *certificate-providing convex-analytic framework* with clear structural meaning.

---

> ### Author Response · Authors · 2025-12-02
>
> #### Comment 2: “There are many known algorithms (e.g., extra-gradient ascent, or ellipsoid) that find Nash equilibria of monotone games (and, more generally, solutions to monotone variational inequalities). Thus, one way to circumvent the hardness of verifying monotonicity is to simply 1) run one of these algorithms, 2) check whether it outputs an equilibrium, and 3) if it does not, assert that the game is non-monotone.”
>
> Response: The reviewer’s second criticism concerns our *SOS-monotone* class, for which verification is efficient (by construction) and equilibrium computation requires *one* step of the hierarchy, giving a polynomial-time algorithm. This is precisely the type of structural improvement the reviewer requested all along. However, they dismiss this by noting that monotone games can be solved by generic methods (e.g., extra-gradient, ellipsoid), and suggest circumventing the hardness of verifying monotonicity by running such an algorithm and checking the output. However, this line of reasoning is, first, not relevant to our contribution and, second, incorrect. Our goal is not to certify monotonicity per se, but to identify a structurally defined subclass of imperfect-recall games that is “easy’’ for SOS methods and for which one has *provably* poly-time, certificate-based equilibrium computation. Checking whether the output of an algorithm is a Nash equilibrium is easy in the non-absentminded case, but is intractable in general. Hence, the reviewer’s suggestion is not a reasonable comparison, and does not give meaningful complexity insights.
>
> #### Comment 3: In their first response that ask for “missing related work [1,2] on team games.” and in their second reply: they note “Regarding the comparison to [1, 2] specifically: Another difference is that they allow mixed strategies, whereas the present paper is about behavioral equilibria. So, except in the single-player timeable case, the settings are rather different. But it is worth noting that in the single-player timeable case, their results seem to dominate those in the present paper, since their upper bounds are always (weakly) better than the trivial that you would get from your result.” The referee insists on comparisons with [1,2], which in our view is somewhat irrelevant.
>
> As the AC can quickly see, papers [1,2] cited by the referee are about two-player zero-sum timeable games, and study team-maximin correlated equilibria. Those results use a “junction tree” type of result by Wainwright and Jordan, to provide an extended formulation for the polytope of correlated strategies, based on a tree decomposition of the dependency graph. They obtain a specific decomposition using the notion of “public states” that leads to a linear formulation, where they give a bound $O(N W^{w+1})$, on the number of nonzero entries of the matrix defining the linear program in [2], where $W$ is the treewidth and $w$ measures the number of uncommon external information. On the other hand, we work with general games (beyond zero-sum/timeable/two-player), behavioral strategies (i.e., not correlated), and our guarantees are in terms of limit/finite convergence of a sequence of SDPs with explicit size.
>
> However, motivated the referee’s comment, we have now obtained a bound on the level of the hierarchy needed for termination that depends on the *treewidth of the constraint graph*, using a junction-tree–type result for the Lasserre hierarchy [3], similar to the result used in [1,2]. We will include this in the revised version of the manuscript.
>
> Finally, for the two-player zero-sum case, we have devised an alternative Moment-SOS hierarchy that converges asymptotically, without the need for a cutting-plane subroutine, simplifying the algorithm in this special case. This is based on the work by F. Bach [4].
>
> *[1] Zhang, Brian Hu, Gabriele Farina, and Tuomas Sandholm. "Team belief DAG: generalizing the sequence form to team games for fast computation of correlated team max-min equilibria via regret minimization." International Conference on Machine Learning. PMLR, 2023.*
>
> *[2] Zhang, Brian Hu, and Tuomas Sandholm. "Team correlated equilibria in zero-sum extensive-form games via tree decompositions." Proceedings of the AAAI Conference on Artificial Intelligence. Vol. 36. No. 5. 2022.*
>
> *[3] Wainwright, Martin J., and Michael I. Jordan. Treewidth-based conditions for exactness of the Sherali-Adams and Lasserre relaxations. Technical Report 671, University of California, Berkeley, 2004.*
>
> *[4] Bach, Francis. "Sum-of-squares relaxations for polynomial min–max problems over simple sets." Mathematical Programming 209.1 (2025): 475-501.*

---

> > ### Author Response · Authors · 2025-12-02
> >
> > As for reviewer kjtu (score 4), his main concern is the scalability of SOS/SDP methods. In the revision, we have added a dedicated paragraph on computational considerations, giving explicit formulas for the number of variables, constraints, and the size of the moment and localizing matrices, etc. This helps readers gauge the realistic scope of our approach with current SDP solvers. To address scalability in Moment-SOS hierarchies, we also discuss several recent works on more scalable SOS/SDP variants (DSOS/SDSOS, low-rank and factored SDP methods, GPU-accelerated solvers, etc.), and our polynomial IREFG formulation naturally fits into these frameworks. This clarifies how one could combine our degree bounds with practically viable solvers on larger benchmarks.
> >
> > Responding to the request, we added a new experiment comparing our SOS method with projected gradient descent (PGD) on a nontrivial single-player IREFG instance. SOS recovers and certifies the global optimum at low degree, whereas PGD frequently gets stuck at suboptimal KKT points, directly demonstrating the qualitative advantage of our framework.
> >
> > Finally, we clarified that the references suggested by the reviewer concern very different settings (two-player zero-sum games or to verify concavity and monotonicity), whereas our work treats general IREFGs with behavioral strategies, underscoring the novelty of our theoretical results.
> >
> > After our response, reviewer kjtu notes that the paper’s main contribution is theoretical and “somewhat outside the scope” of their research, and therefore keeps the score but reduces their confidence. We understand this as a concern about topical fit and personal focus, rather than about the correctness or significance of the results, and hope you will weigh their review in that light alongside the more positive assessments.

---

### Meta-Review · Area_Chair_vnqq · 2025-12-30

**Summary:**

The Reviewers are mixed about this paper. Two of them are proposing acceptance (scores of 8 and 6), while tow of them are proposing rejection (scored of 4 and 2). In particular, Reviewer suQa (score 2) is the most critical about this paper, as they feel like the results are only preliminary, even though the direction taken by the paper seems very promising to the Reviewer. Specifically, Reviewer suQa would have liked more focus on the efficiency of the proposed approach, and a more extensive comparison with two related works, even though they are concerned with the specific setting of team games [1,2]. I tend to agree with Reviewer suQa, and I would like to add that an experimental evaluation of the proposed approach (perhaps on the specific setting of team games) would have made the paper much stringer than the current version, and resolved any potential concern about the actual applicability of the proposed approach. In general, I am not against purely theoretical papers that do not provide an experimental evaluation of the proposed techniques. I thin that this is fine provided that the theoretical results are strong and solid. However, in the case of this paper, I believe that experiments would have been helpful, even if carried on toy/simplified games. Indeed, the theoretical results seem to follow rather smoothly from the literature on the SOS hierarchy, and thus they seem not strong enough to stand out alone. Moreover, historically papers on equilibrium computation in extensive-form games have usually provided some kind of experimental assessment. As such, I believe that the paper in it current state does **not** pass the bar for acceptance at ICLR.

[1] BH Zhang, G Farina, T Sandholm (ICML 2023) "Team Belief DAG: Generalizing the Sequence Form to Team Games for Fast Computation of Correlated Team Max-Min Equilibria via Regret Minimization"

[2] BH Zhang, T Sandholm (AAAI 2022) "Team correlated equilibria in zero-sum extensive-form games via tree decompositions"

**Reviewer Concerns:**

Most of the concerns were raised by Reviewer suQa (score 2). I believe that the Authors have only partially addressed such concerns in the rebuttal. In particular, it remains unclear whether the reposed approach can lead to efficient equilibrium computation algorithms or not. This undermines the strength of this paper. See also the discussion provided in the summary.

**Reviewer Scores:**

Reviewer Zpxn, Score: 8 - I believe that the rebuttal would not have changed the reviewer’s opinion.

Reviewer suQa, Score: 2 - I believe that the rebuttal would not have changed the reviewer’s opinion.

Reviewer hZLc, Score: 6 - I believe that the rebuttal would not have changed the reviewer’s opinion.

Reviewer kjtu, Score: 4 - I believe that the rebuttal would not have changed the reviewer’s opinion.

---

### Decision · Program_Chairs · 2026-01-26

Reject